# RedAHD: Toward End-to-End LLM-Based Automatic Heuristic Design using Reductions

## Abstract

Solving NP-hard combinatorial optimization problems (COPs) (e.g., traveling salesman problems (TSPs) and capacitated vehicle routing problems (CVRPs)) in practice traditionally involves handcrafting heuristics or specifying a search space for finding effective heuristics. The main challenges from these approaches, however, are the sheer amount of domain knowledge and implementation efforts required from human experts. Recently, significant progress has been made to address these challenges, particularly by using large language models (LLMs) to design heuristics within some predetermined generalized algorithmic framework (GAF, e.g., ant colony optimization and guided local search) for building key functions/components (e.g., *a priori* information on how promising it is to include each edge in a solution for TSP and CVRP). Although existing methods leveraging this idea have shown to yield impressive optimization performance, they are far from being end-to-end and still require considerable manual interventions. In this paper, we propose a novel framework, named RedAHD, that enables these LLM-based heuristic design methods to operate without the need of GAFs. More specifically, RedAHD employs LLMs to automate the process of *reduction*, i.e., transforming the COP at hand into similar COPs that are better-understood, from which LLM-based heuristic design methods can design effective heuristics for directly solving the transformed COPs and, in turn, indirectly solving the original COP. Our experimental results, evaluated on six COPs, show that RedAHD is capable of designing heuristics with competitive or improved results over the state-of-the-art methods with minimal human involvement.

## 1 Introduction

Solving NP-hard combinatorial optimization problems (COPs) encountered in real-world applications, such as TSPs (Matai et al., 2010) and CVRPs (Dantzig & Ramser, 1959), traditionally requires extensive domain knowledge and manual efforts from human experts to either design approximation algorithms with provable guarantees or handcraft problem-specific heuristics, with the latter being a more pertinent choice in practice (Desale et al., 2015). In response, automatic heuristic design (AHD), or hyper-heuristics (Burke et al., 2013; Pillay & Qu, 2018), was proposed as a promising alternative, in which the goal is to find the best heuristic among several prespecified options i.e., the heuristic space. Among popular AHD approaches, those employing genetic programming (GP) (Langdon & Poli, 2013), an evolutionary algorithm from machine learning, stands out due to its effectiveness in navigating the heuristic space as well as interpretability (Mei et al., 2022). However, GP-based AHD approaches require a handcrafted set of permissible search operators for generating new heuristics, which can be hard to construct in practice (O'Neill et al., 2010).

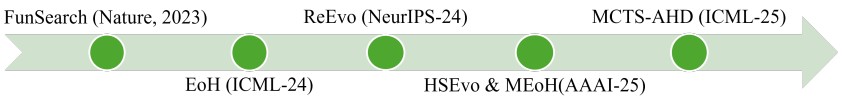

FunSearch (Nature, 2023)  ReEvo (NeurIPS-24)  MCTS-AHD (ICML-25)

EoH (ICML-24)  HSEvo & MEoH(AAAI-25)

Figure 1: Timeline of LLM-EPS methods developed thus far.

**Latest Efforts and Their Limitations.** In recent years, the advent of powerful, readily accessible large language models (LLMs) such as GPT-3.5 and its successors (Brown et al., 2020) has enabled new approaches for AHD (Liu et al., 2024b). Among them, integrating LLMs into an evolutionary computation (EC) procedure for iterative refinement of heuristics, also known as LLM-based evolutionary program search (LLM-EPS) (Liu et al., 2024d; Dat et al., 2025), has attracted increasing attention. As illustrated in Figure 1, in the past two years, multiple works falling into this category have been proposed, each building upon the previous ones to yield incrementally better results. The common idea from these works is to maintain a set of heuristics with good optimization performance on an evaluation dataset of problem instances and iteratively prompt LLMs to generate new heuristics using existing ones as references. LLM-EPS methods can not only design novel, high-quality heuristics but also streamline the implementation process by representing heuristics as LLM-generated code that can be applied to unseen in-distribution (ID) as well as out-of-distribution (OOD) problem instances (Liu et al., 2024a; Ye et al., 2024; Yao et al., 2025; Zheng et al., 2025b). Combined with the current rapid development of LLMs with improved reasoning capabilities (Zheng et al., 2025a), this approach is expected to revolutionize how heuristics for COPs are developed and implemented in the near future (Liu et al., 2024b).

Table 1: GAFs used for the considered COPs in existing LLM-EPS studies. Legends: IC–iterative construction; GLS–guided local search; ACO–ant colony optimization; NCO–neural combinatorial optimization (see Section 4.1 for clarifications on COP acronyms). COPs not considered in the respective studies are shaded.

| COP \ Method | FunSearch 2024 | EoH 2024a | ReEvo 2024 | HSEvo 2025 | MEoH 2025 | MCTS-AHD 2025b | RedAHD (ours) |
|---|---|---|---|---|---|---|---|
| TSP | | GLS | IC, ACO, GLS, NCO | GLS | GLS | IC, ACO | **None** (self-contained) |
| OBPP | IC | IC | | IC | IC | IC | |
| BPP | | | ACO | | | ACO | |
| KP | | | | | | IC | |
| MKP | | | ACO | | | ACO | |
| CVRP | | | ACO, NCO | | | ACO | |

However, despite their advantages over classical AHD approaches, existing LLM-EPS methods are far from being end-to-end (Liu et al., 2024b). That is, they only design heuristics for building key functions/components within some predetermined general algorithmic framework (GAF), such as iterative construction (IC) (Asani et al., 2023), ant colony optimization (ACO) (Dorigo et al., 2007), and guided local search (GLS) (Voudouris et al., 2010), as detailed in Table 1, rather than heuristics for solving COPs directly. When ACO is employed for TSP, for instance, LLM-EPS methods only aim to design heuristics that indicate how promising it is to include each

Table 2: Performance comparison (lower is better) between LLM-EPS methods using IC vs. ACO for TSP (from results in Zheng et al. (2025b)). $n$ is the number of nodes. Note: Results came from different test sets (1,000 and 64 instances for IC and ACO, respectively), hence actual values might vary slightly.

| Setting \ Method | TSP w/ IC | | TSP w/ ACO | |
|---|---|---|---|---|
| | $n=50$ | $n=100$ | $n=50$ | $n=100$ |
| EoH 2024a | 6.394 | 8.894 | 5.828 | 8.263 |
| MCTS-AHD 2025b | 6.225 | 8.684 | 5.801 | 8.179 |

edge in a solution. This heuristic is then used to generate *a priori* information within the ACO framework to better guide the search/foraging behavior of ants. Thus, when applying existing LLM-EPS methods to solve COPs in practice, human users still need to manually specify and design a suitable GAF for directly solving the problem. Employing complex GAFs such as ACO and GLS may yield improved performance over handcrafted heuristics, GP-based AHD methods, and even specialized neural networks (see "NCO" in Appendix A) (Liu et al., 2024a; Ye et al., 2024; Dat et al., 2025; Yao et al., 2025; Zheng et al., 2025b) but also requires domain knowledge and significant implementation efforts, whereas resorting to simple GAFs such as IC may result in subpar performances (see Table 2). In either case, a tailored GAF must be implemented for each COP (see Appendix C for comparison between ACO code for TSP vs. CVRP). Then, individual components for LLM prompting in accordance with the built GAF, e.g., the (sub)problem description, the heuristic description, and the function signature, are carefully designed (see Table S9). Given these limitations, LLM-EPS with enhanced automation warrants more attention to advance the field of AHD (Liu et al., 2024b).

**Our Contributions.** In this paper, we initiate the first attempt toward end-to-end AHD via LLM-EPS. We summarize our contributions as follows:

- We introduce a novel general framework, named **Red**uction-based **A**utomatic **H**euristic **D**esign (RedAHD), that enables existing LLM-EPS methods to function independently without the need

of GAFs. RedAHD operates based on the simple-yet-powerful idea of *reduction* in algorithm design (Crescenzi, 1997) (also formally defined in Section 2), in which a COP of interest is transformed into a similar COP that is better-understood. This process is automated by prompting the LLM to devise a reduction and implement two corresponding functions (as code) that convert instances and solutions of one COP to another. By this means, existing LLM-EPS methods can be utilized to design novel heuristics for directly solving the transformed COP and, in turn, indirectly solving the original COP. RedAHD not only enhances automation in LLM-based AHD, substantially reducing the manual efforts involved, but also potentially brings fresh insights to the COP at hand by uncovering uncharted heuristic space (to be elaborated in Section 3.2) and yields improved optimization performance over state-of-the-art methods.

- We incorporate a mechanism within RedAHD that automatically refines reduction functions (for mapping instances and solutions of one COP to another) whenever the search process stagnates and seemingly converges to local optima (within the landscape defined by the objective function of the COP). This extension, in turn, enables RedAHD to yield good performance even when the initial reductions are not adequately implemented by the LLM.

- We empirically show in our experiments that when integrating the most representative LLM-EPS method, EoH (Liu et al., 2024a), into RedAHD to attempt end-to-end AHD for six COPs, the designed heuristics achieve competitive or better optimization performances compared to existing LLM-EPS methods even when operated under advanced GAFs such as ACO. Moreover, these impressive performances are further improved when we employ (i) a more powerful LLM (o3-mini) or (ii) more sophisticated LLM-EPS methods (ReEvo (Ye et al., 2024) and MEoH (Yao et al., 2025)).

**Outline.** We provide the preliminaries in Section 2. Section 3 describes the proposed RedAHD framework. We evaluate its efficacy through various experiments in Section 4 and conclude our work in Section 5. We defer related work to Appendix A and further discussions (e.g., resource consumption, limitations and future works, and advantage scope of RedAHD) to Appendix D.

## 2 LANGUAGE REDUCTION FOR COMBINATORIAL OPTIMIZATION

In this section, we first revisit the LLM-based AHD task as considered in previous LLM-EPS works, which helps better identify their shared flaw and motivate our approach, then formally define the concept of *reduction* and *language reduction* upon which our framework is built.

Let $A$ be a COP of interest, $x$ be an instance of $A$, $y$ be a feasible solution of $x$, and $h(x) = y$ be a heuristic for $A$. The (supposed) task of LLM-EPS is to search for an optimal heuristic $h^*$ in a heuristic space $H$ (characterized by prior knowledge from LLMs) such that its expected performance on solving $A$ is maximized, i.e.,

$$h^* \in \arg\max_{h \in H} \mathbb{E}_{x \sim \mathcal{D}}\big[q(x, h(x))\big] \tag{1}$$

where $\mathcal{D}$ is an arbitrary distribution over problem instances of $A$ and $q(x, y)$ is the objective function for $A$ (defined in Appendix B for each of our considered COPs). However, existing LLM-EPS methods (Romera-Paredes et al., 2024; Liu et al., 2024a; Ye et al., 2024; Dat et al., 2025; Yao et al., 2025; Zheng et al., 2025b) actually design $h'(x') = y'$, which builds a subroutine within some GAF and hence does not solve the COP on its own. Therefore, in reality, the task of these methods is to search for

$$h^* \in \arg\max_{h' \in H'} \mathbb{E}_{x \sim \mathcal{D}}\Big[q\big(x, g(h'(f(x)))\big)\Big] \tag{2}$$

where $f(x) = x'$ maps an instance of $A$ to an instance of a subproblem $B$ and $g(y') = y$ maps a solution of $B$ to a solution of $A$, both of which are given by the manually specified GAF, and $H'$ is the heuristic space for $B$.

We approach the task by noticing that the tuple $(f, g)$ resembles the following concept of reduction.

**Definition 1 (Reduction (Crescenzi, 1997))** *Let $A$ and $B$ be two COPs. A reduction from $A$ to $B$ is a pair of polynomial-time computable functions $(f, g)$, such that:*

- *$f$ maps an instance $x$ of $A$ into an instance $x'$ of $B$, i.e., $f(x) = x'$.*

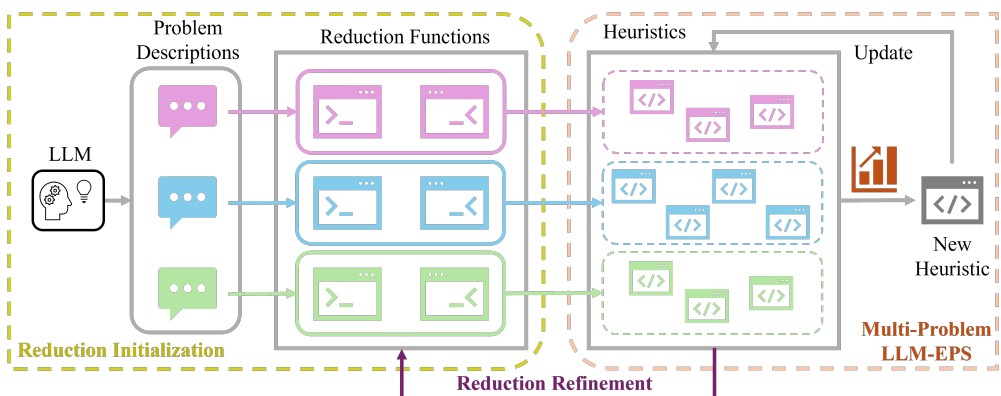

Figure 2: Illustration of RedAHD. First, the designer LLM generates a set of LRs, encoded as two reduction functions (one for mapping instances of $A$ to $B$ and the other for mapping solutions of $B$ to $A$, see Figure S7-center for an example). The LRs are then used to generate a set of heuristics (exemplified in Figure S8) that are iteratively refined using existing LLM-EPS methods, in which offspring heuristics of an LR may be generated using algorithmic ideas from heuristics of any other LRs. When the overall performance of the heuristics associated with an LR stagnates, the LR is automatically refined by the LLM.

- *g maps a solution $y'$ of $B$ to a solution $y$ of $A$, i.e., $g(y') = y$.*

Motivated by this observation, our goal in this paper is thus to automate the design of $f$ and $g$ (Definition 1), which eliminates the need of GAFs and thereby enhance automation in LLM-based AHD. We hereby introduce a novel variant of reduction as follows.

**Definition 2 (Language reduction)** *A language reduction (LR) is an approximate reduction from $A$ to $B$ where $f, g$ are generated by LLMs. The reduction is "approximate" in the sense that $g$ does not necessarily preserve some guarantee of the performance ratio of $y$ with respect to $x$ (Crescenzi, 1997).*

## 3  THE REDAHD FRAMEWORK

In this section, we propose RedAHD, which aims to address the stated flaw of existing LLM-EPS methods via LRs. In essence, RedAHD only takes $A$'s specifications as input and outputs $h^*$ defined in Equation 2 with minimal human involvement. It maintains a set $P$ of $N$ LLM-generated heuristics, denoted as $P = \{h'_1, \ldots, h'_N\}$[1], by adopting some LLM-EPS method to iteratively find heuristics with better objective values subject to a finite set of $D > 0$ problem instances drawn from $\mathcal{D}$. Each heuristic $h'_i \in P$ is associated with an LR $r_j \in R = \{r_1, \ldots, r_M\}$, which transforms $A$ into another COP, $B_j$. The LRs are automatically refined as needed to avoid premature convergence at locally optimal heuristics. Figure 2 illustrates the schematic of RedAHD, which comprises three steps: (i) reduction initialization, (ii) multi-problem LLM-EPS, and (iii) reduction refinement. The following subsections elaborate each step. Our designed prompts are detailed in Appendix D.1.

### 3.1  REDUCTION INITIALIZATION

**LR Representation.**   We start by describing the components to represent an LR, which include:

1. The natural-language problem description of $B$ in a few sentences.

2. The code snippet for implementing $(f, g)$ in accordance with $A$ and $B$'s descriptions. It should follow a predefined format, referred to as "reduction template", so that it can be seamlessly combined with existing LLM-EPS methods. [2]

---

[1] For clarity, $h'$ denotes heuristics for an arbitrary $B$ and $h^{(j)}$ denotes heuristics for a specific $B_j$.

[2] In the experiments, we choose to implement $f$ and $g$ as two Python functions.

3. The code template based on the implemented $(f, g)$, which is used by the employed LLM-EPS method to design $h'$ for $B$. In prior works, this component must be manually designed in accordance with the underlying GAF (see "Function signature" in Table S9).

4. Each LR is assigned a score to quantify its performance on $A$, which is used for selection and stagnation tracking (to be elaborated in Section 3.3). We will define this score shortly.

We provide illustrative examples of LRs in Appendix D.4.

**Candidate LR Generation.** Given $A$'s description, RedAHD first prompts the LLM to provide a list of $M_{init} \geq M$ descriptions for the respective candidate COPs, $\{B_j\}_{j=1}^{M_{init}}$. For each $B_j$, RedAHD generates $(f_j, g_j)$ by prompting the LLM with its description and the reduction template as input, then uses these functions to prompt the LLM again for the code template associated with $B_j$. We do not combine these two sequential calls into one to prevent hallucinations from LLMs (Huang et al., 2025).

**Heuristic Initialization.** We initialize a set of heuristics for each $B_j$, denoted as $P_j$, by providing the LLM with $B_j$'s description and its corresponding code template. Once a heuristic $h_i^{(j)1}$ is generated, its optimization performance, or fitness value, is computed as follows:

$$Q\big(h_i^{(j)}\big) = \frac{1}{D} \sum_{k=1}^{D} q(x^{(k)}, y^{(k)})$$

where $q$ is the objective function for $A$ (e.g., minus tour length for TSP) and $y^{(k)} = g_j\big(h_i^{(j)}(f_j(x^{(k)}))\big)$. We repeat this process $\lceil N/M \rceil$ times to obtain $P_j = \{h_1^{(j)}, \ldots, h_{\lceil N/M \rceil}^{(j)}\}$.

**Selection.** We define the score of an LR, denoted as $s_j$, as the average fitness values of its top-$l$ associated heuristics. After evaluating all $M^{init}$ candidate LRs, we select $M$ LRs with highest scores. Consequently, the initial set of heuristics is $P = \bigcup_{j=1}^{M} P_j$, for a total of at least $N$ heuristics.

Note that for any LR $r_j$, we do not explicitly check the correctness of $(f_j, g_j)$. As long as the solutions $y^{(k)}$ returned from the resulting heuristic $h_i^{(j)}$ are valid (e.g., for TSP, the tour must traverse all nodes without revisiting non-starting nodes) for *all* respective instances $x^{(k)}$, $r_j$ is deemed valid. We elaborate on our strategy to consistently obtain valid LRs in Appendix D.1.

## 3.2 MULTI-PROBLEM LLM-EPS

Once the set of LRs $R$ and the resulting set of heuristics $P$ are initialized, the evolutionary search procedure in RedAHD follows existing LLM-EPS methods, which typically consists of: (i) selecting parent heuristic(s) from $P$ (either randomly or based on $Q$), (ii) applying variation operators on these heuristics via LLM prompting to search for new heuristics in $H'$ (as elaborated in Appendix D.1), and (iii) managing the size of $P$ to be within $N$ by only keeping the fittest heuristics. However, since there are now multiple options for $H'$, it is important to apply these works such that the expanded heuristic space can be efficiently explored without incurring extra costs. Therefore, we extend LLM-EPS methods to *multi-problem settings* where *any* heuristic from $P$, regardless of which COP it is intended to solve, may be indiscriminately selected as parent when designing new heuristics for a given COP. That is, a heuristic $h_i^{(j)}$ for $B_j$ can be used as algorithmic reference to generate offspring heuristics for $B_{j'}, j' \neq j$. The advantages of this technique over designing heuristics for each $B_j$ separately are twofold. First, it prevents situations where one LR performs significantly better than others and hence all heuristics in $P$ are designed for a single COP, making the search for heuristics for other COPs futile. More importantly, it facilitates the discovery of novel heuristics from uncharted heuristic space, which may result in improved performance. Figure 3 illustrates a supporting example for this claim during heuristic design for TSP.

In the following, we describe the multi-problem LLM-EPS procedure using EoH (Liu et al., 2024a) as the reference LLM-EPS method given its close resemblance to traditional evolutionary algorithms and proven significance to the field of LLM-based AHD, but the same concept can be applied to other LLM-EPS methods (as detailed in Appendix D.1 for ReEvo (Ye et al., 2024) and MEoH (Yao et al., 2025)).

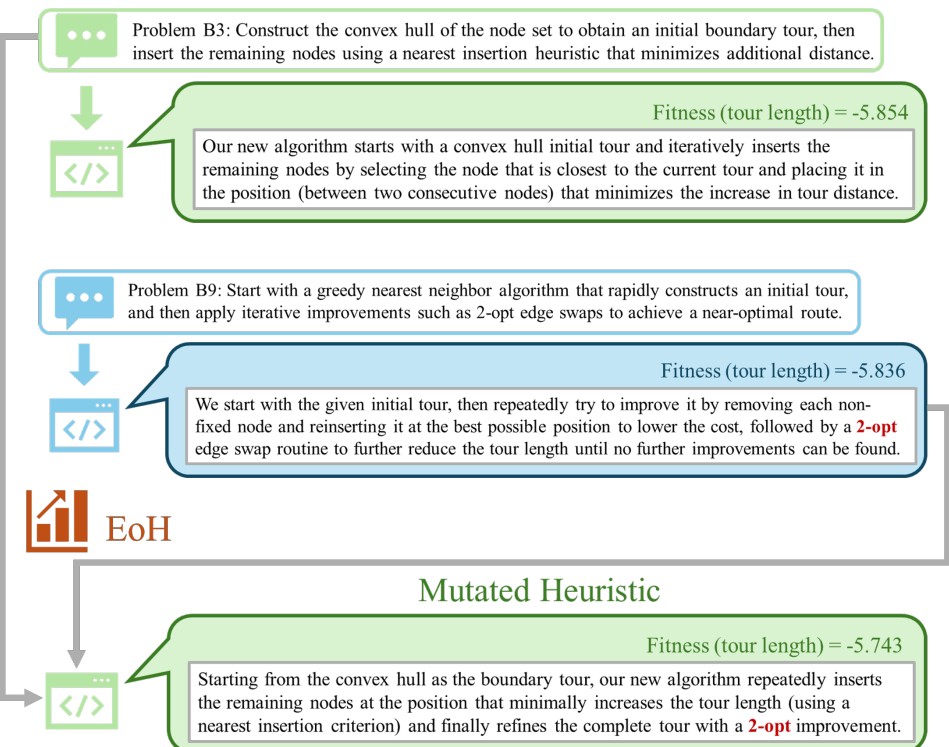

Figure 3: A demonstration of multi-problem LLM-EPS for TSP, in which the parent heuristic (blue) during EoH mutation (Liu et al., 2024a) is not intended to solve the COP at hand ("Problem B3"). As a result, the offspring heuristic for B3 (green) is generated with the novel idea of 2-opt edge swap and hence yields better performance.

**LR ration.** At each iteration or generation in EoH, each variation operator (e.g., crossover and mutation) is applied $N$ times to generate $N$ new heuristics for $B$. In multi-problem LLM-EPS, each variation operator now creates heuristics for different COPs in $\{B_j\}_{j=1}^M$. To maintain the number of newly generated heuristics in a generation, each variation operator is applied to generate only $0 < N_j < N$ heuristics for $B_j$ so that $\sum_{j=1}^M N_j = N$. The exact numbers are determined as follows.

**LR selection.** The number of times $B_j$ is considered for generating new heuristics is determined by sampling $N$ times from $R$ with probability $p_j \propto 1/|s_j|$ if $q(x, y) < 0$ (e.g., TSP) and $p_j \propto s_j$ if $q(x, y) \geq 0$ (e.g., knapsack problems), which resembles the selection method in EoH (Liu et al., 2024a) for selecting parent heuristics. Thus, better-performing reductions are more likely to have larger $N_j$.

### 3.3 REDUCTION REFINEMENT

During evolution, one LR may drastically outperform others (e.g., due to inadequate implementations), securing large ration and in turn monopolizing nearly all heuristics in $P$. Since the search now effectively collapses to typical LLM-EPS, this behavior may lead to premature convergence at local optima (Zheng et al., 2025b). To avoid this, RedAHD automatically refines LRs whenever their score stagnates. In particular, for each $r_j$ when $s_j$ does not improve for $T$ consecutive units of evaluation budget (e.g., number of generations or fitness evaluations), the reduction functions $(f_j, g_j)$ as well as the corresponding code template for $B_j$ are updated by prompting the LLM with both $A$ and $B_j$'s descriptions along with their current version. Once updated, the fitness values of the heuristics associated with $r_j$ are recomputed (through the new $(f_j, g_j)$), which in turn updates $s_j$. RedAHD keeps the update for $r_j$ only if $s_j$ is improved. The exact prompt used is detailed in Appendix D.1.

## 4 EXPERIMENTS

We start this section by describing the experimental settings and the considered baselines in Section 4.1. Section 4.2 presents results on six COPs for evaluating the efficacy of RedAHD. Finally, we provide several ablation studies in Section 4.3 to grasp its individual components' impact on optimization performance. Appendix D includes all implementation details and missing results.

### 4.1 EXPERIMENTAL SETUPS

To our best knowledge, the current state-of-the-art LLM-EPS method is MCTS-AHD (Zheng et al., 2025b). Therefore, we follow their setups whenever possible, including the considered COPs, the evaluation dataset for each COP, and the respective baselines (from handcrafted heuristics, traditional AHD methods, NCO methods, and other LLM-EPS methods). The COPs consist of TSPs, CVRPs, 0/1 knapsack problems (KPs), multiple knapsack problems (MKPs), and online and offline bin packing problems (OBPPs and BPPs, respectively). For RedAHD, we set $M = 3$, $M_{init} = 10$, and $l = 3$. We use EoH (Liu et al., 2024a) as the default LLM-EPS method, in which we use only two variation operators, one for crossover and the other for mutation, instead of five as in the original work (see prompt specifications and our justifications in Appendix D.1). We set $T$, the number of generations in EoH context, to 3. Unless otherwise specified, GPT-4o-mini with temperature fixed at 1 is employed as the designer LLM for generating both LRs and heuristics, with each run of RedAHD repeated three times and we report the average performance of $h^*$.

### 4.2 MAIN RESULTS

Recall that existing LLM-EPS methods necessitate some predetermined GAF to operate. Hence, we compare RedAHD with LLM-EPS methods when integrated within the IC and ACO frameworks.

**Iterative Construction (IC).** This GAF, also known as step-by-step construction, constructs the solution components of a given COP one by one (Asani et al., 2023). By this means, when dealing with TSP for example, LLM-EPS methods only need to design $h'$ that takes the distance matrix and the currently visiting and unvisited nodes as input and returns the next node to visit. It has been considered in all known LLM-EPS works (see Table 1), particularly for TSP, KP, and OBPP. Table 3 shows the performance of RedAHD on these COPs with respect to the baselines. We see that for TSP and KP, RedAHD not only outperforms EoH, the underlying LLM-EPS, but also achieves the best or second best performance on all test sets. For OBPP, despite surpassing the handcrafted heuristics "Best Fit" and "First Fit" in nearly all settings, RedAHD performs rather unremarkably compared to LLM-EPS methods. We attribute this decrease in relative performance to the fact that for OBPP in particular, the additional constraint that each item must be packed sequentially without knowledge on future items greatly restricts $H'$ and hence exploring novel heuristics via the proposed multi-problem LLM-EPS is less beneficial. We show in Section 4.3 that RedAHD can still excel with more capable LLMs.

Table 3: Comparative results for (left) TSP & KP and (right) OBPP when LLM-EPS methods (denoted by an asterisk) employ the IC framework. We use the results reported in Zheng et al. (2025b) for the baselines. $n$ is the number of nodes to visit for TSP and number of items to consider for KP and OBPP, and $W$ is the knapsack capacity for KP and bin size for OBPP. ID settings are underlined while OOD settings are not. The best-performing LLM-based method (with GPT-4o-mini) is shaded, and the overall best method is bolded.

| Problem setting | TSP (Obj. ↓) | | | KP (Obj. ↑) | | |
|---|---|---|---|---|---|---|
| Method | $n$=50 | $n$=100 | $n$=200 | $n$=50 $W$=12.5 | $n$=100 $W$=25 | $n$=200 $W$=25 |
| Greedy 1977 | 6.959 | 9.706 | 13.461 | 19.985 | 40.225 | 57.395 |
| POMO 2020 | **5.697** | **8.001** | 12.897 | 19.612 | 39.676 | 57.271 |
| Funsearch* | 6.357 | 8.850 | 12.372 | 19.988 | 40.227 | 57.398 |
| EoH* | 6.394 | 8.894 | 12.437 | 19.993 | 40.231 | 57.399 |
| MCTS-AHD* | 6.225 | 8.684 | 12.120 | **20.015** | **40.252** | **57.423** |
| RedAHD | 5.767 | 8.006 | **11.164** | 20.006 | 40.248 | 57.416 |

| | OBPP (% optimality gap ↓) | | | | | |
|---|---|---|---|---|---|---|
| $n$ | 1k | 1k | 5k | 5k | 10k | 10k |
| $W$ | 100 | 500 | 100 | 500 | 100 | 500 |
| Best Fit | 4.77 | **0.25** | 4.31 | 0.55 | 4.05 | 0.47 |
| First Fit | 5.02 | **0.25** | 4.65 | 0.55 | 4.36 | 0.50 |
| Funsearch* | 2.45 | 0.66 | 1.30 | **0.25** | 1.05 | **0.21** |
| EoH* | 2.69 | **0.25** | 1.63 | 0.53 | 1.47 | 0.45 |
| ReEvo* | 3.94 | 0.50 | 2.72 | 0.40 | 2.39 | 0.31 |
| HSEvo* | 2.64 | 1.07 | 1.43 | 0.32 | 1.13 | **0.21** |
| MCTS-AHD* | **2.45** | 0.50 | **1.06** | 0.32 | **0.74** | 0.26 |
| RedAHD | 3.78 | 0.99 | 2.82 | 0.55 | 2.61 | 0.40 |

Table 4 shows additional results on TSPLib (Reinelt, 1991), a standard real-world TSP benchmark. Following prior LLM-EPS works (Liu et al., 2024a; Ye et al., 2024; Zheng et al., 2025b), we use the best-performing heuristic among the three runs of RedAHD to report its performance. Since this heuristic (depicted in Appendix D.4 under "TSP") was found to randomly select a starting node,

we run it three times for each TSPLib instance and report the average performance. Clearly, the heuristic from RedAHD outperforms all baselines on every instance, achieving small optimality gap even on very large instances with over 1,500 nodes (shaded in green). On the other hand, LLM-EPS methods often fail to surpass handcrafted heuristics (e.g., Christofides (Christofides, 2022)), particularly on larger instances with a few hundred nodes or more.

Table 4: Results (% optimality gap) on TSPLib when LLM-EPS methods (denoted by an asterisk) employ the IC framework. The number from each instance's name corresponds to the number of nodes. We use the results reported in Duflo et al. (2019); Ye et al. (2024); Zheng et al. (2025b) for the baselines. The best baseline is shaded in gray, and the overall best is bolded.

| TSPLib instance | Christofides 2022 | Greedy 2015 | Nearest insertion | Nearest neighbor 1977 | GPHH-best 2019 | EoH* | ReEvo* | MCTS-AHD* | RedAHD |
|---|---|---|---|---|---|---|---|---|---|
| ts225 | 5.67 | 5.38 | 19.93 | 16.82 | 7.71 | 5.57 | 6.56 | 10.84 | **2.29 ± 0.21** |
| rat99 | 9.43 | 22.30 | 21.05 | 21.79 | 14.09 | 18.78 | 12.41 | 10.46 | **3.47 ± 0.08** |
| rl1889 | 7.60 | 19.44 | 24.34 | 23.74 | 21.09 | - | 17.5 | - | **6.87 ± 0.61** |
| u1817 | 14.15 | 19.78 | 24.07 | 22.20 | 21.21 | - | 16.6 | - | **6.42 ± 0.16** |
| d1655 | 12.65 | 16.31 | 21.35 | 23.86 | 18.69 | - | 17.5 | - | **7.10 ± 0.34** |
| bier127 | 13.03 | 19.50 | 23.05 | 23.25 | 15.64 | 14.05 | 10.79 | 7.56 | **2.32 ± 0.38** |
| lin318 | 13.80 | 18.75 | 24.44 | 25.78 | 14.30 | 14.03 | 16.63 | 14.07 | **5.39 ± 0.17** |
| eil51 | 15.18 | 13.03 | 16.14 | 31.96 | 10.20 | 8.37 | 6.47 | 15.98 | **2.29 ± 0.48** |
| d493 | 9.52 | 16.68 | 20.39 | 24.00 | 15.58 | 12.41 | 13.43 | 11.73 | **3.83 ± 0.28** |
| kroB100 | 9.82 | 16.59 | 21.53 | 26.26 | 14.06 | 13.46 | 12.20 | 11.43 | **2.12 ± 0.84** |
| kroC100 | 9.08 | 12.94 | 24.25 | 25.76 | 16.22 | 16.85 | 15.88 | 8.27 | **3.64 ± 0.24** |
| ch130 | 10.09 | 28.40 | 19.21 | 25.66 | 14.77 | 12.26 | 9.40 | 10.18 | **4.51 ± 0.69** |
| pr299 | 11.23 | 31.42 | 25.05 | 31.42 | 18.24 | 23.58 | 20.63 | 11.23 | **5.45 ± 0.33** |
| fl417 | 15.57 | 12.64 | 25.52 | 32.42 | 22.72 | 20.47 | 19.15 | 10.20 | **3.43 ± 0.52** |
| d657 | 10.41 | 15.76 | 22.84 | 29.74 | 16.30 | - | 16.0 | - | **5.34 ± 0.61** |
| kroA150 | 13.44 | 20.24 | 19.09 | 26.08 | 15.59 | 18.36 | 11.62 | 10.08 | **3.62 ± 0.31** |
| fl1577 | 8.84 | 15.60 | 24.17 | 25.01 | 17.60 | - | 12.1 | - | **3.17 ± 0.38** |
| u724 | 12.04 | 17.20 | 25.58 | 28.45 | 15.54 | - | 16.9 | - | **5.08 ± 0.38** |
| pr264 | 11.28 | 11.89 | 34.28 | 17.87 | 23.96 | 18.03 | 16.78 | 12.27 | **4.97 ± 1.05** |
| pr226 | 14.17 | 21.44 | 28.02 | 24.65 | 15.51 | 19.90 | 18.02 | 7.15 | **1.97 ± 1.13** |
| pr439 | 11.16 | 20.08 | 24.67 | 27.36 | 21.36 | 21.96 | 19.25 | 15.12 | **5.65 ± 0.50** |

**Ant Colony Optimization (ACO).** ACO (Dorigo et al., 2007) is an advanced and well-known GAF that had been applied to more complex COPs such as CVRP and MKP (which are respectively more general COPs than TSP and KP). Under this framework, LLM-EPS methods only need to design heuristics for estimating the potential of each solution component, which is then used as prior information to bias the stochastic sampling of solutions (Ye et al., 2024; Zheng et al., 2025b). Our results for RedAHD on TSP, CVRP, MKP, and BPP with respect to baselines employing ACO are shown in Table 5. Being self-contained, RedAHD still outperforms LLM-EPS methods in nearly all OOD settings and yields competitive performance against them in ID settings. RedAHD also stays competitive against DeepACO (Ye et al., 2023), a representative NCO method based on ACO, in all COPs except CVRP. We show in Section 4.3 that the lackluster performance of RedAHD on CVRP, which we believe to be due to the lack of domain knowledge from GPT-4o-mini, can be significantly improved and even tops DeepACO with more capable LLMs.

Table 5: Comparative results for TSP, CVRP, MKP, and BPP when LLM-EPS methods (denoted by an asterisk) employ the ACO framework. We use the results reported in Zheng et al. (2025b) for the baselines. $n$: number of nodes to visit for TSP and CVRP and number of items to consider for MKP and BPP; $C$: vehicle capacity for CVRP; $m$: number of knapsacks for MKP; $W$: bin size for BPP. ID settings are underlined while OOD settings are not. The best-performing LLM-based method (with GPT-4o-mini) is shaded, and the overall best method is bolded.

| Problem setting / Method | TSP (Obj. ↓) | | CVRP (Obj. ↓) | | MKP (Obj. ↑) | | BPP (Obj. ↓) | |
|---|---|---|---|---|---|---|---|---|
| | $\underline{n{=}50}$ | $n{=}100$ | $\underline{n{=}50}$ $\underline{C{=}50}$ | $n{=}100$ $C{=}50$ | $\underline{n{=}100}$ $\underline{m{=}5}$ | $n{=}200$ $m{=}5$ | $\underline{n{=}500}$ $\underline{W{=}150}$ | $n{=}1,000$ $W{=}150$ |
| ACO 2007 | 5.992 | 8.948 | 11.355 | 18.778 | 22.738 | 40.672 | 208.828 | 417.938 |
| DeepACO 2023 | 5.842 | 8.282 | **8.888** | **14.932** | 23.093 | 41.988 | **203.125** | **405.172** |
| EoH* | 5.828 | 8.263 | 9.359 | 15.681 | 23.139 | 41.994 | 204.646 | 408.599 |
| ReEvo* | 5.856 | 8.340 | 9.327 | 16.092 | 23.245 | 42.416 | 206.693 | 413.510 |
| MCTS-AHD* | **5.801** | 8.179 | 9.286 | 15.782 | **23.269** | 42.498 | 204.094 | 407.323 |
| RedAHD | 5.819 | **8.039** | 9.826 | 15.726 | 23.164 | **42.682** | 203.344 | 405.359 |

## 4.3 ABLATION STUDIES

**Reduction Refinement.** In our experiments, for $T = 3$, the reduction refinement step in RedAHD was called at least once up to three times. We validate the necessity of this step by rerunning the experiments in Table 5 without it. As shown in Table 6, RedAHD exhibits a decrease in perfor-

mance across all COPs and barely surpasses EoH. This performance drop is likely due to premature convergence at local optima during search as discussed in Section 3.3.

Table 6: Ablation of the reduction refinement step. Results from EoH in Table 5 are used as references.

| Problem setting Method | TSP (Obj. ↓) | | CVRP (Obj. ↓) | | MKP (Obj. ↑) | | BPP (Obj. ↓) | |
|---|---|---|---|---|---|---|---|---|
| | $n$=50 | $n$=100 | $n$=50 $C$=50 | $n$=100 $C$=50 | $n$=100 $m$=5 | $n$=200 $m$=5 | $n$=500 $W$=150 | $n$=1,000 $W$=150 |
| EoH* | 5.828 | 8.263 | **9.359** | **15.681** | 23.139 | 41.994 | 204.646 | 408.599 |
| RedAHD (w/o reduction refinement) | 5.847 | 8.322 | 10.218 | 16.175 | 23.126 | 41.978 | 204.561 | 407.639 |
| RedAHD | **5.819** | **8.039** | 9.826 | 15.726 | **23.164** | **42.682** | **203.344** | **405.359** |

**The Designer LLM.** The impressive performance from RedAHD across multiple COPs up to this point was achieved using GPT-4o-mini, a lightweight general-purpose LLM that had been shown to be poor at algorithmic reasoning (Yang et al., 2025). Therefore, we should expect RedAHD to improve when more capable LLMs, particularly reasoning models such as o3-mini, are employed. Table 7 verifies our claim, where the originally unremarkable performance from RedAHD on OBPP and CVRP is significantly improved and even surpasses the best baseline on multiple settings. Notably, for the OOD setting of CVRP ($N = 100$ and $C = 50$), RedAHD yields objective values even better than those returned from OR-Tools, an optimization library dedicated for vehicle routing problems (Furnon & Perron).

Table 7: Ablation of the designer LLM. Truncated results from Tables 3 and 5 are used as references.

| OBPP (% optimality gap ↓) | | | | | | |
|---|---|---|---|---|---|---|
| $n$ (number of items) | 1k | 1k | 5k | 5k | 10k | 10k |
| $W$ (bin capacity) | 100 | 500 | 100 | 500 | 100 | 500 |
| Best baseline | **2.45** | 0.25 | **1.06** | 0.25 | **0.74** | 0.21 |
| EoH* | 2.69 | 0.25 | 1.63 | 0.53 | 1.47 | 0.45 |
| RedAHD (GPT-4o-mini) | 3.78 | 0.99 | 2.82 | 0.55 | 2.61 | 0.40 |
| RedAHD (o3-mini) | 3.13 | **0.00** | 2.33 | 0.30 | 2.02 | **0.20** |

| CVRP (Obj. ↓) | | |
|---|---|---|
| $n$ (number of nodes) | 50 | 100 |
| $C$ (vehicle capacity) | 50 | 50 |
| OR-Tools (Furnon & Perron) | **8.314** | 13.948 |
| Best baseline (DeepACO) | 8.888 | 14.932 |
| EoH* | 9.359 | 15.681 |
| RedAHD (GPT-4o-mini) | 9.826 | 15.726 |
| RedAHD (o3-mini) | 8.348 | **13.516** |

**The LLM-EPS Method.** We demonstrate that RedAHD can work with LLM-EPS methods other than EoH, namely ReEvo (Ye et al., 2024) and MEoH (Yao et al., 2025). As shown in Table 8, RedAHD improves the performance of the corresponding LLM-EPS methods even without the need of GAFs. In particular, RedAHD[EoH] and RedAHD[ReEvo] respectively outperform EoH and ReEvo, where the latter two operate under the ACO framework. Moreover, as LLM-EPS methods improve, exemplified here by MEoH (which extends EoH to multi-objective heuristic search with runtime as the additional fitness criterion), RedAHD may yield further improvement, now outperforming the best baseline in the ID setting of TSP ($N = 50$). This result verifies the applicability of our proposed framework in the emerging field of LLM-based AHD.

Table 8: Ablation of the underlying LLM-EPS method. Truncated results from Table 5 are used as references. RedAHD[EoH] is RedAHD reported in earlier results. For RedAHD[MEoH], which also optimizes runtime, we report the average performance from heuristics that yield the best objective values.

| TSP (Obj. ↓) | | |
|---|---|---|
| $n$ (number of nodes) | 50 | 100 |
| Best baseline | 5.801 | 8.179 |
| EoH* | 5.828 | 8.263 |
| ReEvo* | 5.856 | 8.340 |
| RedAHD[EoH] | 5.819 | 8.039 |
| RedAHD[ReEvo] | 5.835 | 8.251 |
| RedAHD[MEoH] | **5.730** | **7.883** |

## 5 CONCLUSION

In this paper, we propose RedAHD, the first framework toward end-to-end automatic design of heuristics with LLMs. RedAHD leverages the concept of reduction for enabling contemporary LLM-EPS methods to operate without the need of GAFs, which significantly reduces manual efforts from human designers. Furthermore, RedAHD facilitates the discovery of novel heuristics from uncharted heuristic space, resulting in improved optimization performance over state-of-the-art methods. As the capabilities of LLMs and LLM-EPS methods continue to grow, we envision the efficacy of RedAHD in solving COPs would be more evident.

## 6 REPRODUCIBILITY STATEMENT

We refer readers to Section 4.1 as well as Appendix D.1 for complete details on reproducing our results. We also include the code for our work in the supplemental material.

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

## A  RELATED WORK

**Automatic Heuristic Design (AHD).**  The field of AHD, or hyper-heuristics (Pillay & Qu, 2018), aims to provide more generalized approaches for solving COPs via selecting the best-performing heuristic from a predefined set (Drake et al., 2020) or generating new heuristics through the combination of simpler heuristic components (Duflo et al., 2019; Zhao et al., 2023). By this means, human experts are only required to specify the heuristic space rather than handcrafting heuristics from scratch. However, traditional AHD approaches such as those employing GP (Langdon & Poli, 2013) necessitate substantial domain knowledge and implementation efforts (Pillay & Qu, 2018; O'Neill et al., 2010).

**LLMs for AHD.**  Recent advances in LLMs have enabled new approaches for AHD. (Please refer to the latest survey by Liu et al. (2024b)) for a comprehensive review.) Since standalone LLMs with prompt engineering are arguably incapable of producing novel algorithmic ideas beyond their encoded knowledge (Mahowald et al., 2024), most active research in this area focuses on integrating LLMs into an evolutionary computation (EC) procedure to iteratively refine a set of heuristics. EC is a generic optimization principle inspired by natural evolution (Bäck et al., 1997; Eiben & Smith, 2015). Its idea involves iteratively improving a set of candidate solutions through score-based selection (i.e., identifying the "fittest" candidate solutions subject to a so-called fitness function such as the optimality gap) and stochastic variation operators (e.g., crossover and mutation among the fittest candidate solutions as inspired by biological evolution). In recent years, LLMs have been employed via prompt engineering to emulate these variation operators (Lehman et al., 2023; Meyerson et al., 2024; Lange et al., 2024), with already widespread applications in code generation (Hemberg et al., 2024), text generation (Guo et al., 2024), planning (Kambhampati et al., 2024), as well as AHD, known in the literature as LLM-based evolutionary program search (LLM-EPS) (Liu et al., 2024d; Dat et al., 2025). Representative LLM-EPS methods include FunSearch (Romera-Paredes et al., 2024), EoH (Liu et al., 2024a), ReEvo (Ye et al., 2024), HSEvo (Dat et al., 2025), MeoH (Yao et al.,

2025), and most recently MCTS-AHD (Zheng et al., 2025b) (Figure 1). Despite generally outperforming handcrafted heuristics and GP-based AHD methods while reducing manual interventions, as mentioned in Section 1, they rely on some predetermined GAF such as IC and ACO to operate, which still involves domain knowledge and implementation efforts from human users, and hence are far from being end-to-end. In response, our work enables existing LLM-EPS methods to circumvent this limitation and potentially improves their performance in the process.

**Neural Combinatorial Optimization (NCO).** NCO is an end-to-end AHD approach that employs neural networks to search for the optimal parameter settings within a parameterized heuristic space (Bengio et al., 2021; Yang & Whinston, 2023). Despite not requiring domain knowledge and being applicable to multiple COPs (Chen et al., 2023; Ma et al., 2023), compared to LLM-EPS methods, they are resource-intensive (Kwon et al., 2020), hard to implement (Zheng et al., 2024), and may yield subpar results in various experimental settings (Liu et al., 2024a; Ye et al., 2024; Zheng et al., 2025b), being outperformed by the state-of-the-art LLM-EPS method, MCTS-AHD, even under the simple IC framework when solving TSP and the 0/1 knapsack problem (Zheng et al., 2025b) for instance. (Please refer to existing LLM-EPS works for a more comprehensive comparison with NCO methods.)

## B    CONSIDERED COPs

In this appendix, we introduce the considered COPs and define the objective function for each ($q$ in Equations 1 and 2). We follow the problem definitions and setups from Zheng et al. (2025b) (which followed Ye et al. (2024)). TSP, CVRP, BPP, and OBPP are minimization problems while KP and MKP are maximization problems.

**Traveling Salesman Problem (TSP).** TSP aims to find the shortest path to visit each of the $n$ nodes once and return to the starting node. Each TSP instance contains the Euclidean distance matrix $\boldsymbol{D}$ where $d_{ij}$ denotes the cost between node $i$ and $j$. The solution of TSP is a permutation of all node indices $\boldsymbol{s} = (s_1, s_2, \ldots, s_n)$. Thus, the (negated) objective function is

$$-\left(\sum_{t=1}^{n-1} d_{s_t, s_{t+1}} + d_{s_n, s_1}\right).$$

**Capacitated Vehicle Routing Problem (CVRP).** CVRP aims to plan several capacity-constrained vehicles starting at and returning to a depot, meeting the demands of multiple customers, and minimizing the total travel distance. Each CVRP instance contains a depot (the 0-th node) and $n$ customers. Let $\boldsymbol{D}$ be the Euclidean distance matrix. The (negated) objective function is

$$
\begin{aligned}
&-\sum_{j=1}^{q} C\left(\boldsymbol{\rho}^j\right), \\
&C\left(\boldsymbol{\rho}^j\right) = \sum_{t=0}^{|\boldsymbol{\rho}^j|-1} d_{\rho_t^j, \rho_{t+1}^j}^j + d_{\rho_{n_j}^j, \rho_0^j}^j, \\
\text{s.t.} \quad &0 \le \delta_i \le C, \quad \sum_{i \in \boldsymbol{\rho}^j} \delta_i \le C, \quad i \in \{1, \ldots, n\}, j \in \{1, \ldots, q\},
\end{aligned}
$$

where $\boldsymbol{s}$ is a solution representing the complete route of vehicles and consists of $q$ sub-routes $\boldsymbol{s} = \{\boldsymbol{\rho}^1, \boldsymbol{\rho}^2, \ldots, \boldsymbol{\rho}^q\}$. Each sub-route $\boldsymbol{\rho}^j = (\boldsymbol{\rho}_1^j, \ldots, \boldsymbol{\rho}_{n_j}^j)$, $j \in \{1, \ldots, q\}$ starts from the depot $s_0$ and goes back to $s_0$; $n_j$ represents the number of customer nodes in it such that $n = \sum_{j=1}^{q} n_j$. $\delta_i$ denotes the demand of node $i$, and $C$ denotes the capacity of the vehicles.

**(Offline) Bin Packing Problem (BPP).** BPP aims to place a set of $n$ items with different sizes into as few bins as possible, each of which has capacity of $W$. The solution of BPP is $\boldsymbol{s} = \{\boldsymbol{s}^1, \boldsymbol{s}^2, \ldots, \boldsymbol{s}^K\}$ where $\boldsymbol{s}^i$ is the set of item indices for the $i$-th bin and $K$ is the number of bins used. The (negated) objective function is

$$
\begin{aligned}
&-K, \\
\text{s.t.} \quad &\sum_{j \in \boldsymbol{s}^i} w_j \le W, \quad i \in \{1, \ldots, K\}.
\end{aligned}
$$

**Online Bin Packing Problem (OBPP).** OBPP additionally requires making an immediate decision on which bin to place once a new item arrives, without any information on future items. The objective function is similar to BPP.

**0/1 Knapsack Problem (KP).**    KP aims to pack items of maximum total value to a knapsack with capacity $W$. Each of the $n$ available items can only be picked once. The solution of KP is the set of indices of the selected items $s \subseteq \{1, 2, \ldots, n\}$. Let $w_j$ and $v_j$ be the weight and value of item $j$, respectively. The objective function is

$$\sum_{j \in s} v_j,$$
$$\text{s.t.} \quad \sum_{j \in s} w_j \leq W.$$

**Multiple Knapsack Problem (MKP).**    MKP extends KP to $m > 1$ knapsacks. The solution of MKP is now $s = \{s^1, s^2, \ldots, s^m\}$ where $s^i$ is the set of indices of the selected items for the $i$-th knapsack. The objective function is

$$\sum_{i=1}^{m} \sum_{j \in s^i} v_j,$$
$$\text{s.t.} \quad \sum_{j \in s^i} w_j \leq W_i, \quad i \in \{1, \ldots, m\}.$$

## C   HOW ACO IS EMPLOYED IN PRIOR LLM-EPS WORKS.

As described in Zheng et al. (2025b), ACO is an evolutionary algorithm inspired by the behavior of ants to find the shortest route between their colony and food sources (Dorigo et al., 2007).

ACO records a pheromone matrix $\tau$ and a heuristic matrix $\eta$. Each item $\tau_{ij}$ in $\tau$ indicates the priority of including an edge $(i, j)$ in a solution. The pheromone trails are iteratively updated based on the quality of the solutions found, encouraging future ants to follow better paths. The heuristic information on each edge, i.e., $\eta_{ij}$, is a problem-specific measure that indicates the immediate benefit of choosing a particular path. For solving TSP with ACO, for example, $\eta_{ij}$ is often set to be the inverse of the distance between cities $i$ and $j$, i.e., $\eta_{ij} = 1/d_{ij}$. In response, LLM-EPS methods aim to design a more effective heuristic matrix $\eta$ based on the problem-specific inputs.

Given $\eta$, the virtual ants then construct solutions by moving from node to node, probabilistically choosing the next node based on a combination of pheromone and heuristic information. After all the ants have constructed their solutions, the pheromone levels update. An ACO iteration typically involves solution construction, optional local search, and pheromone update. By iteratively applying these steps, ACO algorithms can effectively explore the solution space and converge toward optimal or near-optimal solutions for COPs.

**Implementation.**   The following listings respectively show the Python implementation of ACO for TSP and CVRP in both ReEvo (Ye et al., 2024) and MCTS-AHD (Zheng et al., 2025b). Albeit using the same GAF, there are substantial differences between the two pieces of code, which means significant manual efforts are necessary when adopting ACO (and other GAFs in general) for a particular COP.

```python
import torch
from torch.distributions import Categorical

class ACO():

    def __init__(self,
                 distances,
                 heuristic,
                 n_ants=30,
                 decay=0.9,
                 alpha=1,
                 beta=1,
                 device='cpu'
                 ):

        self.problem_size = len(distances)
        self.distances = torch.tensor(distances, device=device) if not isinstance(distances, torch.Tensor) else
            distances
        self.n_ants = n_ants
        self.decay = decay
        self.alpha = alpha
        self.beta = beta

        self.pheromone = torch.ones_like(self.distances)
        self.heuristic = torch.tensor(heuristic, device=device) if not isinstance(heuristic, torch.Tensor) else
            heuristic

        self.shortest_path = None
        self.lowest_cost = float('inf')

        self.device = device

    @torch.no_grad()
    def run(self, n_iterations):
        for _ in range(n_iterations):
            paths = self.gen_path(require_prob=False)
            costs = self.gen_path_costs(paths)

            best_cost, best_idx = costs.min(dim=0)
            if best_cost < self.lowest_cost:
                self.shortest_path = paths[:, best_idx]
                self.lowest_cost = best_cost

            self.update_pheronome(paths, costs)

        return self.lowest_cost

    @torch.no_grad()
    def update_pheronome(self, paths, costs):
        '''
        Args:
            paths: torch tensor with shape (problem_size, n_ants)
            costs: torch tensor with shape (n_ants,)
```

```python
        '''
        self.pheromone = self.pheromone * self.decay
        for i in range(self.n_ants):
            path = paths[:, i]
            cost = costs[i]
            self.pheromone[path, torch.roll(path, shifts=1)] += 1.0/cost
            self.pheromone[torch.roll(path, shifts=1), path] += 1.0/cost

    @torch.no_grad()
    def gen_path_costs(self, paths):
        '''
        Args:
            paths: torch tensor with shape (problem_size, n_ants)
        Returns:
            Lengths of paths: torch tensor with shape (n_ants,)
        '''
        assert paths.shape == (self.problem_size, self.n_ants)
        u = paths.T # shape: (n_ants, problem_size)
        v = torch.roll(u, shifts=1, dims=1) # shape: (n_ants, problem_size)
        assert (self.distances[u, v] > 0).all()
        return torch.sum(self.distances[u, v], dim=1)

    def gen_path(self, require_prob=False):
        '''
        Tour contruction for all ants
        Returns:
            paths: torch tensor with shape (problem_size, n_ants), paths[:, i] is the constructed tour of the
                    ith ant
            log_probs: torch tensor with shape (problem_size, n_ants), log_probs[i, j] is the log_prob of the
                    ith action of the jth ant
        '''
        start = torch.randint(low=0, high=self.problem_size, size=(self.n_ants,), device=self.device)
        mask = torch.ones(size=(self.n_ants, self.problem_size), device=self.device)
        mask[torch.arange(self.n_ants, device=self.device), start] = 0

        paths_list = [] # paths_list[i] is the ith move (tensor) for all ants
        paths_list.append(start)

        log_probs_list = [] # log_probs_list[i] is the ith log_prob (tensor) for all ants' actions

        prev = start
        for _ in range(self.problem_size-1):
            actions, log_probs = self.pick_move(prev, mask, require_prob)
            paths_list.append(actions)
            if require_prob:
                log_probs_list.append(log_probs)
                mask = mask.clone()
            prev = actions
            mask[torch.arange(self.n_ants, device=self.device), actions] = 0

        if require_prob:
            return torch.stack(paths_list), torch.stack(log_probs_list)
        else:
            return torch.stack(paths_list)

    def pick_move(self, prev, mask, require_prob):
        '''
        Args:
            prev: tensor with shape (n_ants,), previous nodes for all ants
            mask: bool tensor with shape (n_ants, p_size), masks (0) for the visited cities
        '''
        pheromone = self.pheromone[prev] # shape: (n_ants, p_size)
        heuristic = self.heuristic[prev] # shape: (n_ants, p_size)
        dist = ((pheromone ** self.alpha) * (heuristic ** self.beta) * mask) # shape: (n_ants, p_size)
        dist = Categorical(dist)
        actions = dist.sample() # shape: (n_ants,)
        log_probs = dist.log_prob(actions) if require_prob else None # shape: (n_ants,)
        return actions, log_probs
```

Listing 1: Implementation of the ACO framework for TSP in ReEvo (Ye et al., 2024) and MCTS-AHD (Zheng et al., 2025b).

```python
import torch
from torch.distributions import Categorical
import random
import itertools
import numpy as np

class ACO():
    def __init__(self, # 0: depot
                 distances, # (n, n)
                 demand, # (n, )
                 heuristic, # (n, n)
                 capacity,
                 n_ants=30,
                 decay=0.9,
                 alpha=1,
                 beta=1,
                 device='cpu',
```

```python
            ):

    self.problem_size = len(distances)
    self.distances = torch.tensor(distances, device=device) if not isinstance(distances, torch.Tensor) else
        distances
    self.demand = torch.tensor(demand, device=device) if not isinstance(demand, torch.Tensor) else demand
    self.capacity = capacity

    self.n_ants = n_ants
    self.decay = decay
    self.alpha = alpha
    self.beta = beta

    self.pheromone = torch.ones_like(self.distances)
    self.heuristic = torch.tensor(heuristic, device=device) if not isinstance(heuristic, torch.Tensor) else
        heuristic

    self.shortest_path = None
    self.lowest_cost = float('inf')

    self.device = device

@torch.no_grad()
def run(self, n_iterations):
    for _ in range(n_iterations):
        paths = self.gen_path()
        costs = self.gen_path_costs(paths)

        best_cost, best_idx = costs.min(dim=0)
        if best_cost < self.lowest_cost:
            self.shortest_path = paths[:, best_idx]
            self.lowest_cost = best_cost

        self.update_pheronome(paths, costs)

    return self.lowest_cost

@torch.no_grad()
def update_pheronome(self, paths, costs):
    '''
    Args:
        paths: torch tensor with shape (problem_size, n_ants)
        costs: torch tensor with shape (n_ants,)
    '''
    self.pheromone = self.pheromone * self.decay
    for i in range(self.n_ants):
        path = paths[:, i]
        cost = costs[i]
        self.pheromone[path[:-1], torch.roll(path, shifts=-1)[:-1]] += 1.0/cost
    self.pheromone[self.pheromone < 1e-10] = 1e-10

@torch.no_grad()
def gen_path_costs(self, paths):
    u = paths.permute(1, 0) # shape: (n_ants, max_seq_len)
    v = torch.roll(u, shifts=-1, dims=1)
    return torch.sum(self.distances[u[:, :-1], v[:, :-1]], dim=1)

def gen_path(self):
    actions = torch.zeros((self.n_ants,), dtype=torch.long, device=self.device)
    visit_mask = torch.ones(size=(self.n_ants, self.problem_size), device=self.device)
    visit_mask = self.update_visit_mask(visit_mask, actions)
    used_capacity = torch.zeros(size=(self.n_ants,), device=self.device)

    used_capacity, capacity_mask = self.update_capacity_mask(actions, used_capacity)

    paths_list = [actions] # paths_list[i] is the ith move (tensor) for all ants

    done = self.check_done(visit_mask, actions)
    while not done:
        actions = self.pick_move(actions, visit_mask, capacity_mask)
        paths_list.append(actions)
        visit_mask = self.update_visit_mask(visit_mask, actions)
        used_capacity, capacity_mask = self.update_capacity_mask(actions, used_capacity)
        done = self.check_done(visit_mask, actions)

    return torch.stack(paths_list)

def pick_move(self, prev, visit_mask, capacity_mask):
    pheromone = self.pheromone[prev] # shape: (n_ants, p_size)
    heuristic = self.heuristic[prev] # shape: (n_ants, p_size)
    dist = ((pheromone ** self.alpha) * (heuristic ** self.beta) * visit_mask * capacity_mask) # shape:
        (n_ants, p_size)
    dist = Categorical(dist)
    actions = dist.sample() # shape: (n_ants,)
    return actions

def update_visit_mask(self, visit_mask, actions):
    visit_mask[torch.arange(self.n_ants, device=self.device), actions] = 0
    visit_mask[:, 0] = 1 # depot can be revisited with one exception
    visit_mask[(actions==0) * (visit_mask[:, 1:]!=0).any(dim=1), 0] = 0 # one exception is here
    return visit_mask
```

```
def update_capacity_mask(self, cur_nodes, used_capacity):
    '''
    Args:
        cur_nodes: shape (n_ants, )
        used_capacity: shape (n_ants, )
        capacity_mask: shape (n_ants, p_size)
    Returns:
        ant_capacity: updated capacity
        capacity_mask: updated mask
    '''
    capacity_mask = torch.ones(size=(self.n_ants, self.problem_size), device=self.device)
    # update capacity
    used_capacity[cur_nodes==0] = 0
    used_capacity = used_capacity + self.demand[cur_nodes]
    # update capacity_mask
    remaining_capacity = self.capacity - used_capacity # (n_ants,)
    remaining_capacity_repeat = remaining_capacity.unsqueeze(-1).repeat(1, self.problem_size) # (n_ants,
        p_size)
    demand_repeat = self.demand.unsqueeze(0).repeat(self.n_ants, 1) # (n_ants, p_size)
    capacity_mask[demand_repeat > remaining_capacity_repeat] = 0

    return used_capacity, capacity_mask

def check_done(self, visit_mask, actions):
    return (visit_mask[:, 1:] == 0).all() and (actions == 0).all()
```

Listing 2: Implementation of the ACO framework for CVRP in ReEvo (Ye et al., 2024) and MCTS-AHD (Zheng et al., 2025b).

**Manually Designed Prompts for TSP and CVRP in Existing Works.** In prior LLM-EPS works, prompt components for calling LLMs must be designed in accordance with the employed GAF, rather than the COP at hand. In Table S9, we compare these components when ACO is employed for TSP vs. CVRP.

Table S9: Prompt components used in ReEvo (Ye et al., 2024) and MCTS-AHD (Zheng et al., 2025b) under the ACO framework.

TSP

| Prompt component | Specification |
| --- | --- |
| Problem description | Solving Traveling Salesman Problem (TSP) via stochastic solution sampling following "heuristics". TSP requires finding the shortest path that visits all given nodes and returns to the starting node. |
| Heuristic description | The 'heuristics' function takes as input a distance matrix, and returns prior indicators of how promising it is to include each edge in a solution. The return is of the same shape as the input. |
| Function signature | `def heuristics(distance_matrix: np.ndarray) -> np.ndarray:` |

CVRP

| Prompt component | Specification |
| --- | --- |
| Problem description | Solving Capacitated Vehicle Routing Problem (CVRP) via stochastic solution sampling. CVRP requires finding the shortest path that visits all given nodes and returns to the starting node. Each node has a demand and each vehicle has a capacity. The total demand of the nodes visited by a vehicle cannot exceed the vehicle capacity. When the total demand exceeds the vehicle capacity, the vehicle must return to the starting node. |
| Heuristic description | The 'heuristics' function takes as input a distance matrix (shape: n by n), Euclidean coordinates of nodes (shape: n by 2), a vector of customer demands (shape: n), and the integer capacity of vehicle capacity. It returns prior indicators of how promising it is to include each edge in a solution. The return is of the same shape as the distance matrix. The depot node is indexed by 0. |
| Function signature | `def heuristics(distance_matrix: np.ndarray, coordinates: np.ndarray, demands: np.ndarray, capacity: int) -> np.ndarray:` |

# D ADDITIONAL EXPERIMENTS AND DISCUSSIONS

## D.1 COMPLETE IMPLEMENTATION DETAILS

All experiments were conducted under Ubuntu 20.04 on a Linux virtual machine equipped with NVIDIA GeForce RTX 3050 Ti GPU and 12th Gen Intel(R) Core(TM) i7-12700H CPU @2.3GHz. The code for our implementation in Python 3.10 is uploaded as supplementary material.

We adopt the experimental setups from MCTS-AHD (Zheng et al., 2025b), the state-of-the-art LLM-EPS method, to better gauge the efficacy of RedAHD in solving COPs. For the evaluation datasets, we use their publicly available data[3] during both training and testing for all considered COPs.

**RedAHD Settings.** We set $M = 3$, $M_{init} = 10$, and $l = 3$. Prompts for LR generation and refinement are specified in Figures S5 and S4, respectively. The running time of each heuristic on the evaluation dataset for any COP is limited to 60 seconds. We use EoH (Liu et al., 2024a) as the default LLM-EPS method, in which we use only two variation operators, one for crossover and the other for mutation, instead of five as in the original work (see prompt specifications and our justifications in EoH Settings). We set $T$, the number of generations in EoH context, to 3. Additionally, during population management at early stages of evolution, we do not discard heuristics with identical objective values if they are from different LRs. This ensures every LR has sufficient heuristics (at least $l$) for obtaining a valid score. Unless otherwise specified, GPT-4o-mini with temperature fixed at 1 is employed as the designer LLM for generating both LRs and heuristics, with each run of RedAHD repeated three times and we report the average performance of $h^*$.

Because RedAHD is self-contained, solution checks are necessary to ensure the validity of the generated heuristics and LRs. That is, during fitness evaluation, we check the solution of each instance as follows:

- **TSP.** All nodes must be visited exactly once.
- **CVRP.** (i) Each customer from a sub-route must be visited exactly once; (ii) sum of demands from customers served by a sub-route must not exceed the vehicle capacity; (iii) all customers must be visited exactly once.
- **BPP.** All items must be packed in one of the bins without exceeding the capacity of any bin.
- **OBPP.** The selected bin must have sufficient capacity for packing the current item.
- **KP.** All selected items must be unique and their total weight must not exceed the knapsack capacity.
- **MKP.** All selected items across all knapsacks must be unique and the total weight of the items in any knapsack must not exceed its capacity.

> **Prompt for Reduction Refinement**
>
> Problem A: [Problem A Description]
>
> I want to transform Problem A into another problem, Problem B, that can be solved efficiently while still providing near-optimal solutions to Problem A. I have one option for Problem B as follows:
>
> Problem description: [Problem B Description]
>
> Please help me modify the following code for transforming Problem A to Problem B and vice versa while remaining as efficient as possible.
>
> Code:
> [Reduction Functions]
>
> Do not give additional explanations.

Figure S4: Prompts used for reduction refinement in RedAHD as described in Section 3.3.

---

[3]https://github.com/zz1358m/MCTS-AHD-master/tree/main

**Prompt for Candidate LR Initialization**

Problem A: [Problem Description]

I want to transform Problem A into another problem, Problem B, that can be solved efficiently while still providing near-optimal solutions to Problem A. Please help me devise $M_{init}$ different Problem B's. Describe each Problem B in a sentence or two (without mentioning Problem A) and enclose it inside a double brace as follows:

{{Problem B1 involves ...}}
{{Problem B2 involves ...}}
...

Do not give additional explanations.

**Prompt for Generating Reduction Functions**

Problem A: [Problem A Description]

Problem B: [Problem B Description]

Implement 2 Python functions for transforming Problem A into Problem B using the following templates:

[Reduction Template]

Only provide me the code without any further explanations.

**Prompt for Code Template Generation**

I have the following code for transforming a Problem A into a simplified Problem B and vice versa.

Code:
[Reduction Functions]

Using this information, fill in the blanks of the following Python function template.

Code template:
[Heuristic Template]

First, determine <INPUT_B> from output of 'convert_input_A_to_B()'. Then, determine <SOLUTION_B> from 'solution_B' variable in 'convert_solution_B_to_A()'. Finally, complete the docstring at <ARGS> and <RETURNS> with as detailed type hints as possible. Do not attempt to solve the problem directly and do not give additional explanations.

```python
import numpy as np
from typing import Tuple

def convert_input_A_to_B(coord_matrix, distance_matrix):
    ''' Convert input of Problem A into input of Problem B
    Args:
    [ARGS]

    Returns:
    input_B: A tuple storing the corresponding input of
        Problem B.
    '''

    # Placeholder (replace with your actual implementation)
    input_B = ...

    return input_B

def convert_solution_B_to_A(solution_B):
    ''' Convert solution of Problem B into solution of
        Problem A
    Args:
    solution_B: The output of Problem B.

    Returns:
    [RETURN]
    '''

    # Placeholder (replace with your actual implementation)
    [PLACEHOLDER]
```

```python
from typing import Tuple

def solve_B(<INPUT_B>):
    '''
    Args:
    <ARGS>

    Returns:
    <RETURNS>
    '''

    return <SOLUTION_B>
```

Figure S5: Prompts used for candidate LR generation in RedAHD as described in Section 3.1. The chronological order for LLM prompting is (top) ▶ (center left) ▶ (center right). The (bottom left) code snippet is the "Reduction Template", where [ARGS], [RETURN], [PLACEHOLDER] are COP-specific and detailed in Table S11. The (bottom right) code snippet is the "Heuristic Template".

**EoH Settings.**  Following Zheng et al. (2025b), the number of generations in EoH is set to 20 and the population size $N$ is set to 20 for CVRP, BPP, OBPP, MKP and 10 for TSP and KP. EoH utilizes five variation operators in total, two for crossover (E1 and E2) and three for mutation (M1, M2, M3). RedAHD only uses E2 and M1 from EoH (see Figure S6, bottom) since we actually observed reduced optimization performance when either E1, M2, or M3 is included (and significant increase in runtime and API cost). In particular, we notice the heuristics generated by E1 are often erroneous (due to e.g., code errors or returning invalid solutions). We attribute this behavior to the fact that E1 prompts the designer LLM to generate a completely new heuristic from the provided ones, which might not be well-suited for multi-problem LLM-EPS within RedAHD that already enables ample exploration of novel heuristics.

---

**Prompt for Initialization**

[Problem Description]

I need help design a novel efficient algorithm to solve the problem. First, describe your algorithm and main steps in one sentence. The description must be inside a brace. Next, implement it in Python using the following template:

[Code Template]

Do not give additional explanations.

---

**Prompt for Crossover/Exploration**

[Problem Description]

I have 2 existing algorithms with their codes as follows:
No. 1 algorithm and the corresponding code are:
[Algorithm 1 Description]
[Code 1]

No. 2 algorithm and the corresponding code are:
[Algorithm 2 Description]
[Code 2]

Please help me create a new algorithm that has a totally different form from the given ones but can be motivated from them. First, identify the common backbone idea in the provided algorithms. Secondly, based on the backbone idea describe your new algorithm in one sentence. The description must be inside a brace. Thirdly, implement it in Python using the following template:

[Code Template]

Do not give additional explanations.

---

**Prompt for Mutation/Modification**

[Problem Description]

I have one algorithm with its code as follows.
Algorithm description: [Algorithm Description]
Code: [Code]

Please help me create a new algorithm that has a different form but can be a modified version of the provided algorithm. First, describe your new algorithm and main steps in one sentence. The description must be inside a brace. Next, implement it in Python using the following template:

[Code Template]

Do not give additional explanations.

---

Figure S6: Prompts used for initialization, exploration, and modification in EoH. "Problem Description" and "Code Template" are with respect to $B$ from the LLM-generated LR (see Figure S7 for an example).

**MEoH Settings.**  MEoH (Yao et al., 2025) extends EoH to additionally consider runtime during fitness evaluation via the proposed dominance-dissimilarity mechanism for multi-objective parent selection and population management. We similarly use two variation operators as detailed in EoH Settings. Importantly, each LR now records two scores, one with respect to the objective value and the other with respect to runtime. The latter is defined as the average runtime of its top-$l$ associated heuristics with best objective values. For stagnation tracking, if neither score improves after $T$, then the reduction refinement step is invoked for the LR.

**ReEvo Settings.**  ReEvo (Ye et al., 2024) incorporates reflections into the evolutionary search by prompting the designer LLM to analyze and revise previously generated heuristics. We make the following changes to ReEvo. During parent selection, LR ration is similarly applied to maintain the number of generated offspring heuristics from the two crossover and mutation operators. Short- and long-term reflections are performed for each LR. For short-term reflection, the problem description is with respect to $B_j$. Importantly, in accordance with our proposed multi-problem LLM-EPS in RedAHD, the two provided heuristics can be from $B_{j'}, j' \neq j$.

**COP-Specific Prompts.**    Tables S10 and S11 respectively list the problem descriptions and reduction templates used in prompts. To facilitate the generation of valid LRs that can generalize to OOD instances (i.e., instances with smaller or larger sizes than what originally encountered during training), when specifying problem descriptions and reduction templates, we ensure all COP parameters are abstracted, such as 'N' instead of the actual number of nodes in training instances.

Table S10: Problem descriptions used in prompts.

| COP | Problem description |
|-----|---------------------|
| TSP | Given a set of N nodes with their 2D coordinates, the problem involves finding the shortest route that visits each node exactly once and returns to the starting node. |
| CVRP | Given a set of N customers and a fleet of vehicles with limited capacity, the problem involves finding a corresponding set of optimal routes to deliver goods to all customers. |
| BPP | Given a set of N items with different sizes and some bins each with fixed capacity, the problem involves placing each item inside one of the bins in a way that minimizes the number of bins used without exceeding the bin capacity. |
| OBPP | Given an item with certain size and a set of M bins each with finite capacity, the problem involves finding a priority score for each bin. The bin with the highest priority score will be selected for inserting the item. |
| KP | Given a set of N items with weights and values, the problem involves selecting a subset of items that maximizes the total value without exceeding the knapsack's weight capacity. |
| MKP | Given a set of N items with values and M-dimensional weights, the problem involves selecting a subset of items to maximize the total value without exceeding the multi-dimensional maximum weight constraints. |

Table S11: COP-specific components for reduction templates.

| Component | Specification |
|-----------|---------------|
| **TSP** | |
| ARGS | ```'''```
```coord_matrix (np.ndarray): A Nx2 matrix storing the 2D coordinates of the nodes.```
```distance_matrix (np.ndarray): A NxN matrix where the entry at i-th row and j-th column```
```    (or vice versa) stores the Euclidean distance between nodes i and j.```
```'''``` |
| RETURN | ```'''```
```route: A Numpy 1D array of length N storing the unique node IDs to visit in order.```
```'''``` |
| PLACEHOLDER | ```route = ...```

```return route``` |
| **CVRP** | |
| ARGS | ```'''```
```coord_matrix (np.ndarray): A (N+1)-by-2 matrix storing the Euclidean coordinates of the```
```    depot (first row) and the customers.```
```distance_matrix (np.ndarray): A (N+1)-by-(N+1) distance matrix.```
```demands (np.ndarray): An array of length N+1 storing the customer demands, where the```
```    first entry is 0 (placeholder for the depot).```
```capacity (int): The capacity of each vehicle for satisfying the customer demands.```
```'''``` |
| RETURN | ```'''```
```routes (List[List[int]]): A list of routes; each route is represented as a list of```
```    unique customer indices (1 to N) to visit in order, subject to the capacity```
```    constraint.```
```'''``` |
| PLACEHOLDER | ```routes = []```
```...```

```return routes``` |
| **BPP** | |
| ARGS | ```'''```
```items (np.ndarray): Array of length N storing the item sizes to be considered in exact```
```    order.```
```bins (np.ndarray): Array of capacities for each bin.```
```'''``` |
| RETURN | ```'''```
```packed_bins (np.ndarray): Array of remaining capacities for each bin after packing all```
```    items.```
```'''``` |
| PLACEHOLDER | ```packed_bins = ...```
```...```

```return packed_bins``` |
| **OBPP** | |
| ARGS | ```'''```
```item_size (float): Size of the item to be added to one of the bins.```
```bin_caps (np.ndarray): Array of length M storing capacities of each bin.```
```'''``` |
| RETURN | ```'''```
```scores (np.ndarray): Array of priority scores for the bins.```
```'''``` |

```
PLACEHOLDER   scores = ...
              ...

              return scores
```
| KP |
|---|

```
ARGS          '''
              weights (np.ndarray): A 1D float array of length {problem_size} storing the item
                  weights.
              values (np.ndarray): A 1D float array of length {problem_size} storing the associated
                  item values.
              capacity (float): The weight capacity of the knapsack.
              '''
```
```
RETURN        '''
              items: A list storing the indices of selected items subject to the capacity constraint.
              '''
```
```
PLACEHOLDER   items = []
              ...

              return items
```
| MKP |
|---|

```
ARGS          '''
              values (np.ndarray): A 1D float array of length N storing the item values.
              weights (np.ndarray): A (M x N) float matrix storing the multi-dimensional weights,
                  where each row is associated with a constraint.
              constraints (np.ndarray): A 1D float array of length M storing weight constraints.
              '''
```
```
RETURN        '''
              items: A list storing the indices of selected items subject to the weight constraints.
              '''
```
```
PLACEHOLDER   items = []
              ...

              return items
```

## D.2 ADDITIONAL RESULTS

**Ablation of Multi-Problem LLM-EPS.** We validate the necessity of multi-problem LLM-EPS by limiting $M$ to 1, which means the search now becomes typical LLM-EPS. As shown in Table S12, compared to $M = 3$ as we did throughout our previous experiments, there is a significant decrease in optimization performance across all COPs. This result supports our claims of multi-problem LLM-EPS advantages as discussed in Section 3.2.

Table S12: Ablation of the proposed multi-problem LLM-EPS. "RedAHD ($M = 3$)" is RedAHD reported in earlier results. Results from EoH in Table 5 are used as references.

| Problem setting / Method | TSP (Obj. ↓) | | CVRP (Obj. ↓) | | MKP (Obj. ↑) | | BPP (Obj. ↓) | |
|---|---|---|---|---|---|---|---|---|
| | $n$=50 | $n$=100 | $n$=50 $C$=50 | $n$=100 $C$=50 | $n$=100 $m$=5 | $n$=200 $m$=5 | $n$=500 $W$=150 | $n$=1,000 $W$=150 |
| EoH* | 5.828 | 8.263 | **9.359** | **15.681** | 23.139 | 41.994 | 204.646 | 408.599 |
| RedAHD ($M = 1$) | 5.931 | 8.479 | 10.327 | 16.252 | 22.925 | 41.569 | 205.983 | 411.428 |
| RedAHD ($M = 3$, without multi-problem LLM-EPS) | 5.943 | 8.602 | 10.537 | 16.985 | 22.916 | 41.497 | 206.013 | 412.220 |
| RedAHD ($M = 3$) | **5.819** | **8.039** | 9.826 | 15.726 | **23.164** | **42.682** | **203.344** | **405.359** |

**Sensitivity to Initial LRs.** We investigate the sensitivity of RedAHD performance to the quality of the initial pool of LRs. Table S13 compares the average test performance of all heuristics in the initial generation (second column) to the test performance of the best heuristic in the final generation (last column). Even though the quality of the initial LRs differs across runs, the final performance of RedAHD remains consistent.

Table S13: RedAHD performance on TSP ($n = 50$) across five independent runs (lower values are better).

| Run | Quality of initial LRs | Final performance |
|---|---|---|
| 1 | 6.831 | 5.784 |
| 2 | 6.378 | 5.761 |
| 3 | 6.494 | 5.775 |
| 4 | 6.796 | 5.770 |
| 5 | 6.592 | 5.766 |

**RedAHD for Solving Flow Shop Scheduling Problems (FSSPs).** FSSP (Emmons & Vairak-takis, 2012) is a complex COP considered in EoH (Liu et al., 2024a) that concerns scheduling $n$ jobs on $m$ machines, where each job involves $m$ operations that must be performed in a pre-determined order on the respective machine. The objective is to minimize the total schedule length,

known as the makespan. We apply RedAHD to FSSP by adopting the same experimental setups from Liu et al. (2024a) (with consistent evaluation datasets and EoH settings) while keeping the same RedAHD settings detailed in Appendix D.1. Table S14 shows that RedAHD attains second-best optimization performance in nearly all settings, surpassing classical FSSP heuristics and even dedicated deep learning solvers. Note that in the original paper, Liu et al. (2024a) employed GLS (Voudouris et al., 2010) as the GAF for EoH, which yields the overall best performance in exchange for additional manual efforts. For completeness, we also run EoH under the IC framework, which seeks to design a heuristic for selecting the next operation given the current status of each machine and job and the set of feasible operations. When employing this simple GAF, we see a substantial drop in EoH performance. Our results thus demonstrate that even without relying on GAFs, RedAHD can effectively handle complex COPs beyond vehicle routing (TSP, CVRP) and packing problems (OBPP, BPP, KP, MKP).

Table S14: Comparative results for FSSP captured by the average (%) gap with respect to the best known makespan (lower is better). We use the results reported in Liu et al. (2024a) for the baselines other than EoH-IC. The best and second-best methods are respectively bolded and shaded.

| | | n20m10 | n20m20 | n50m10 | n50m20 | n100m10 | n100m20 |
|---|---|---|---|---|---|---|---|
| Handcrafted | GUPTA 1971 | 23.42 | 21.79 | 20.11 | 22.78 | 15.03 | 21.00 |
| | CDS 1970 | 12.87 | 10.35 | 12.72 | 15.03 | 9.36 | 13.55 |
| | NEH 1983 | 4.05 | 3.06 | 3.47 | 5.48 | 2.07 | 3.58 |
| | NEHFF 2014 | 4.15 | 2.72 | 3.62 | 5.10 | 1.88 | 3.73 |
| Deep learning | PFSPNet 2021 | 14.78 | 14.69 | 11.95 | 16.95 | 8.21 | 16.47 |
| | PFSPNet_NEH 2021 | 4.04 | 2.96 | 3.48 | 5.05 | 1.72 | 3.56 |
| LLM-EPS | EoH-GLS | **0.30** | **0.10** | **0.19** | **0.60** | **0.14** | **0.41** |
| | EoH-IC | 3.76 | 50.6 | 14.2 | 10.4 | 12.5 | 21.2 |
| | RedAHD | 3.27 | 2.40 | 3.32 | 4.10 | 1.78 | 3.00 |

### D.3 BLACK-BOX SETTINGS

Black-box settings were considered in ReEvo (Ye et al., 2024) and MCTS-AHD (Zheng et al., 2025b), in which all information regarding the COP (e.g., the problem description, the heuristic description, and the function signature in accordance with the designed GAF as shown in Table S9) is not provided. The goal is to fairly evaluate the efficacy of LLM-EPS methods in designing effective heuristics for a wide range of COPs, rather than merely retrieving code tailored to prominent COPs from LLMs' parameterized knowledge. Since RedAHD solves the COP at hand directly without the need of GAFs, the proposed black-box settings in these works are not applicable to RedAHD.

To address the stated concerns regarding mere code retrieval by LLMs, in every considered COP, we do not mention its commonly known name in the problem description (see Table S10. That is, we do not refer to the COPs plainly as e.g., "the traveling salesman problem", but rather vaguely "the problem". Moreover, when prompting the designer LLM for generating heuristics for $B$, we do not leak any information on $A$ (which also helps mitigate hallucinations (Huang et al., 2025)). By this means, **RedAHD already operates under black-box settings by default**.

### D.4 EXAMPLES OF DESIGNED LRs AND HEURISTICS FROM REDAHD

**MKP.** Figures S7 and S8 respectively show an example of the designed LR and the corresponding heuristic for MKP.

```python
import numpy as np
from typing import Tuple, List

def convert_input_A_to_B(values: np.ndarray, weights: np.ndarray, constraints: np.ndarray) ->
        Tuple[np.ndarray, np.ndarray, np.ndarray]:
    ''' Convert input of Problem A into input of Problem B
    Args:
    values (np.ndarray): A 1D float array of length N storing the item values.
    weights (np.ndarray): A (M x N) float matrix storing the multi-dimensional weights, where each row is
        associated with a constraint.
    constraints (np.ndarray): A 1D float array of length M storing weight constraints.

    Returns:
    input_B: A tuple storing the corresponding input of Problem B.
    '''
    # Calculate value-to-weight ratios for each item
    ratios = values / np.sqrt(np.sum(np.square(weights), axis=0)) # Changed to root of sum of squares for
        better ratio
    input_B = (values, weights, constraints, ratios)

    return input_B

def convert_solution_B_to_A(solution_B: List[int]) -> List[int]:
    ''' Convert solution of Problem B into solution of Problem A
    Args:
    solution_B: The output of Problem B, which contains indices of selected items.

    Returns:
    items: A list storing the indices of selected items subject to the weight constraints.
    '''
    items = list(solution_B)
    return items
```

```python
def convert_input_A_to_B(values, weights, constraints):
    ''' Convert input of Problem A into input of Problem B
    Args:
    values (np.ndarray): A 1D float array of length N storing the item values.
    weights (np.ndarray): A (M x N) float matrix storing the multi-dimensional weights, where each row is
        associated with a constraint.
    constraints (np.ndarray): A 1D float array of length M storing weight constraints.

    Returns:
    input_B: A tuple storing the corresponding input of Problem B.
    '''
    # Calculate value-to-weight ratios for each item
    ratios = values / np.sum(weights, axis=0)
    input_B = (values, weights, constraints, ratios)

    return input_B

def convert_solution_B_to_A(solution_B):
    ''' Convert solution of Problem B into solution of Problem A
    Args:
    solution_B: The output of Problem B.

    Returns:
    items: A list storing the indices of selected items subject to the weight constraints.
    '''
    selected_items = solution_B # Assuming solution_B contains the indices of selected items
    items = list(selected_items)

    return items
```

Figure S7: Designed LR for MKP using RedAHD. (Top) Problem description of $B$, (center) refined and (bottom) original implementation of $(f, g)$ for transforming MKP to $B$.

```
from typing import Tuple, List

def solve_B(input_B: Tuple[np.ndarray, np.ndarray, np.ndarray, np.ndarray]) -> List[int]:
    '''
    Args:
    input_B (Tuple[np.ndarray, np.ndarray, np.ndarray, np.ndarray]): A tuple containing:
        - values (np.ndarray): A 1D float array of length N storing the item values.
        - weights (np.ndarray): A (M x N) float matrix storing the multi-dimensional weights, where each row is
            associated with a constraint.
        - constraints (np.ndarray): A 1D float array of length M storing weight constraints.
        - ratios (np.ndarray): A 1D float array of length N storing the value-to-weight ratios for each item.

    Returns:
    List[int]: A list storing the indices of selected items subject to the weight constraints.
    '''

    return solution_B
```

Figure S7 (cont.): Code template for solving $B$.

**Problem Description**

Problem B1 involves selecting a subset of N items such that the total value is maximized while ensuring that the total weights in each of the M dimensions do not exceed specific limits, using a greedy heuristic approach based on value-to-weight ratios.

**Heuristic Description**

A new algorithm that selects items iteratively, calculating the best score considering both value and the remaining capacity left in multi-dimensional space, while simultaneously updating the constraints as items are selected.

```
from typing import Tuple, List
import numpy as np

def solve_B(input_B: Tuple[np.ndarray, np.ndarray, np.ndarray, np.ndarray]) -> List[int]:
    values, weights, constraints, ratios = input_B
    M, N = weights.shape
    selected_items = []
    total_weights = np.zeros(M)

    # Calculate the remaining capacity to define the score more effectively
    remaining_capacity = constraints.copy()

    while True:
        best_score = -np.inf
        best_item = -1
        for idx in range(N):
            if idx in selected_items:
                continue
            item_weight = weights[:, idx]

            if all(total_weights + item_weight <= constraints):
                # Calculate new score based on value and remaining capacity
                score = values[idx] / (np.sum(item_weight / remaining_capacity) + 1e-9) # Avoid division by zero
                if score > best_score:
                    best_score = score
                    best_item = idx

        if best_item == -1: # No feasible item can be added
            break

        selected_items.append(best_item)
        total_weights += weights[:, best_item]
        remaining_capacity -= weights[:, best_item]

    return selected_items
```

Figure S8: Designed heuristic using the LR for MKP in Figure S7.

**TSP.** Figure S9 shows the learned heuristic whose performance was reported in Table 4.

> **Problem Description**
>
> Problem B7 involves applying a nearest neighbor heuristic starting from each node and selecting the route with the least total distance as a near-optimal solution.

> **Heuristic Description**
>
> Repeatedly select a random starting point and build a route while ensuring that each node is added without forming a local loop, then refine the tour using a hill-climbing strategy to find better routes.

```python
from typing import Tuple
import numpy as np

def solve_B(input_B: Tuple[np.ndarray]) -> np.ndarray:
    '''
    Args:
    input_B (Tuple[np.ndarray]): A tuple containing a single np.ndarray, which is a NxN matrix where the entry
                        at i-th row and j-th column (or vice versa) stores the Euclidean distance between
                        nodes i and j.

    Returns:
    np.ndarray: A 1D array of length N containing the unique node IDs to visit in order, representing the
            solution
                of Problem B converted to the corresponding solution of Problem A.
    '''
    distances = input_B[0]
    num_nodes = distances.shape[0]
    best_route = []
    best_length = float('inf')

    # {Repeatedly select a random starting point and build a route while ensuring that each node is added
            without forming a local loop,
    # then refine the tour using a hill-climbing strategy to find better routes.}
    for _ in range(10): # Number of iterations to random start points
        start_node = np.random.randint(num_nodes)
        visited = [start_node]
        unvisited = set(range(num_nodes))
        unvisited.remove(start_node)

        while unvisited:
            current_node = visited[-1]
            next_node = min(unvisited, key=lambda x: distances[current_node][x])
            visited.append(next_node)
            unvisited.remove(next_node)

        # Hill-climbing optimization
        improvement = True
        while improvement:
            improvement = False
            for i in range(len(visited)):
                for j in range(i + 2, len(visited)):
                    if j == len(visited) - 1 and i == 0: # Skip the final edge to start
                        continue
                    current_cost = (distances[visited[i]][visited[(i + 1) % len(visited)]] +
                                distances[visited[j]][visited[(j + 1) % len(visited)]])
                    new_cost = (distances[visited[i]][visited[j]] +
                                distances[visited[(i + 1) % len(visited)]][visited[(j + 1) % len(visited)]])

                    if new_cost < current_cost:
                        visited[i + 1:j + 1] = reversed(visited[i + 1:j + 1])
                        improvement = True
                        break
                if improvement:
                    break

        current_length = sum(distances[visited[k]][visited[(k + 1) % len(visited)]] for k in
                range(len(visited)))

        if current_length < best_length:
            best_length = current_length
            best_route = visited

    return np.array(best_route)
```

Figure S9: Designed heuristic for TSP using RedAHD.

## D.5 RESOURCE CONSUMPTION

Using our employed settings (detailed in Appendix D.1), RedAHD costs at most $0.3 (GPT-4o-mini) or $2 (o3-mini) and 1.5 hour to complete training. The authors of ReEvo argued that the efficiency benchmarking for LLM-EPS methods should prioritize the number of fitness evaluations over the number of LLM calls (Section 7 in Ye et al. (2024)). Additionally, the work of MCTS-AHD, which is the latest LLM-EPS method at the time of submission, also adopted this benchmarking scheme (Appendix D in Zheng et al. (2025b)). Therefore, we estimate the number of fitness evaluations as follows. Since we mainly consider EoH in this work (with two variation operators), RedAHD requires at least $(M_{init} \times \lceil N/M \rceil) + (T_{gen} \times N \times 2) = (10 \times \lceil 20/3 \rceil) + (20 \times 20 \times 2) = 870$ fitness evaluations, where $T_{gen}$ is the number of generations. Each LR refinement additionally requires $N_j < N$ evaluations. Overall, RedAHD needs no more than 1,000 evaluations, which is similar to or lower than the budget used in prior LLM-EPS works (Liu et al., 2024a; Zheng et al., 2025b). In general, the actual costs from running RedAHD naturally follow the costs associated with existing LLM-EPS methods. There are no extra incurred costs during the evolutionary search given our proposed LR ration technique (Section 3.2). The additional number of LLM queries is negligible: $1 + 2 \times M_{init}$ for reduction initialization and 1 for each refinement of an LR.

## D.6 LIMITATIONS AND FUTURE WORKS

First, while RedAHD significantly reduces human involvement in LLM-based AHD for solving COPs, it is yet to be fully end-to-end. That is, RedAHD minimally requires the manual design of (i) prompts for candidate LR generation (Figure S5), which include COP-specific components, and (ii) solution checks during fitness evaluation (bullet points in RedAHD Settings). We believe works from the burgeoning field of LLM planning (Tantakoun et al., 2025; Wei et al., 2025) could be employed to achieve full automation.

Second, effective reductions from RedAHD rely on the encoded knowledge of the designer LLM. In the absence of relevant domain knowledge, it is possible that the designed LRs are trivial. That is, $f$ would simply return the input for $A$ and $g$ would return the raw output of the designed heuristics. In other words, the heuristics would be designed for solving $A$ directly without any reduction involved, which likely results in subpar optimization performance. Thus, when encountering such behavior in practice, we recommend using more capable LLM models during the reduction initialization step (which should be inexpensive as stated in Appendix D.5), before switching back to more budget-friendly models during the remaining steps of RedAHD.

Lastly, as observed in our results with OBPP in Table 3, RedAHD might not perform as well on COPs with restricted heuristic space. To investigate this observation, we further experiment RedAHD on the vehicle routing problem with time windows (VRPTW) (Kallehauge et al., 2005), which is a more restrictive variant of CVRP where each customer $i$ is only available during a specific time window $[t_i^{start}, t_i^{end}]$. VRPTW is a challenging COP with no feasibility guarantees even with the IC framework, and hence employing LLM-EPS methods requires even more manual efforts.[4] Using the library developed by Liu et al. (2024c) to generate 64 50-node training instances and 64 50-node test instances, we run RedAHD (following the aforementioned setups with GPT-4o-mini) and notice the solutions returned from the designed heuristics are not consistently valid. As shown in Figure S10, even after 1000 fitness evaluations, we observe violations of time window constraints in more than 40% of the test instances. When we relax the constraints of VRPTW by lifting $t_i^{start}$, which allows vehicles to fulfill customers' demands early, there is a significant decrease in the percentage of violations, down to approximately 25%. To ensure validity of the generated heuristics across all instances in this challenging setting, given RedAHD's flexibility, a potential workaround could be adopting the structure of existing GAFs from LLM-EPS methods (which guarantee feasible solutions[5]) during fitness evaluation, at the cost of reduced automation. Future studies may consult the EC literature (Wu et al., 2024) for devising better ways of navigating the search within the confined heuristic space $H'$.

---

[4]All prior LLM-EPS methods did not consider this COP, though we are aware of a recent Python library for LLM-based AHD (Liu et al., 2024c) that implements a variant of the IC framework specifically for VRPTW.

[5]As an example, please refer to the tailored IC framework for VRPTW in the library from Liu et al. (2024c).

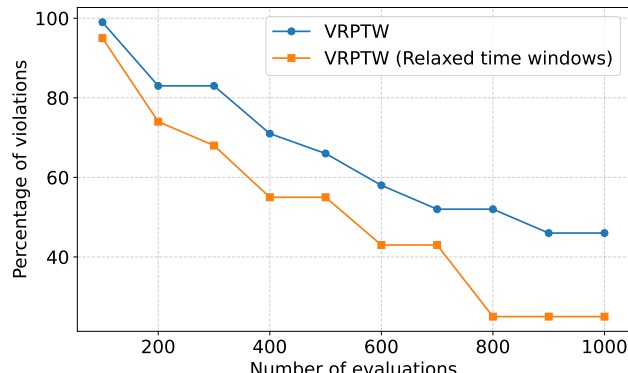

Figure S10: Percentage of test instances for VRPTW in which the heuristics designed by RedAHD violate time window constraints. The orange line considers a relaxed version of VRPTW where vehicles can arrive and serve customers early (i.e., the time windows are $[0, t_i^{end}]$ for all customers $i$).

## D.7 THE ADVANTAGE SCOPE OF REDAHD

Knowing the strengths and current limitations of RedAHD, we summarize application scenarios where our framework would excel.

- *COPs with large heuristic space.* Using reductions and multi-problem LLM-EPS, RedAHD benefits from the many alternatives for $H'$, which indicates RedAHD is more suitable for COPs with less restrictive heuristic space $H$ (e.g., BPP). In practice, the application scenarios could involve designing effective heuristics for a newly formulated COP with moderate constraints.

- *Well-studied COPs in need of performance enhancement.* Since the quality of the designed LRs relies on the domain knowledge of LLMs, we believe RedAHD would perform particularly well on application scenarios where the problem of interest can be formulated as classical COPs (e.g., TSP and MKP) and available off-the-shelf methods (e.g., approximation algorithms and handcrafted heuristics) yield unsatisfactory optimization performance.

