# OpenReview forum: "RedAHD: Toward End-to-End LLM-Based Automatic Heuristic Design using Reductions"
_ICLR.cc/2026/Conference — Submitted to ICLR 2026_

### Official Review · Reviewer_3HNV · 2025-10-29

**Soundness:** 3
**Presentation:** 3
**Contribution:** 3
**Rating:** 8
**Confidence:** 4

**Summary:**

This paper introduces RedAHD, a novel framework for automatic heuristic design (AHD) that aims to make the process more end-to-end by leveraging Large Language Models (LLMs). The central idea is to address a key limitation of prior LLM-based evolutionary program search (LLM-EPS) methods, which rely on manually specified generalized algorithmic frameworks (GAFs) like Ant Colony Optimization or Guided Local Search. RedAHD automates this by using an LLM to perform reductions: transforming the combinatorial optimization problem (COP) at hand into a similar, better-understood problem. This allows an underlying LLM-EPS method to design heuristics for the transformed problem directly, thereby indirectly solving the original one. The framework includes a "multi-problem" evolutionary search, where ideas can be exchanged between heuristics for different reductions, and a "reduction refinement" mechanism to improve reductions when the search stagnates. The authors demonstrate through extensive experiments on six COPs that RedAHD, without needing a GAF, can design heuristics that achieve competitive or state-of-the-art performance compared to methods that do.

**Strengths:**

1. The core idea of using an LLM to learn problem reductions is a paradigm shift for LLM-based AHD, moving beyond heuristic generation to problem transformation.
2. The work directly and effectively tackles the reliance on manually-designed GAFs, a major bottleneck in previous state-of-the-art LLM-EPS methods.
3. The framework is rigorously tested on six different COPs, consistently demonstrating competitive or superior performance against strong baselines on both synthetic and real-world (TSPLib) benchmarks.
4. The paper is well-written and well-organized, making complex ideas accessible. The figures, tables, and extensive appendices contribute to a high-quality and reproducible research artifact.

**Weaknesses:**

1. While RedAHD removes the need for a GAF, it introduces its own set of hyperparameters (e.g., $M, M_{init}, l, T$). The paper lacks a sensitivity analysis for these parameters, making it unclear how crucial their specific tuning is.
2. Limited interpretability of generated reductions—no systematic analysis of what reductions the LLM tends to produce.
3. Some experimental setups and the choice of hyperparameters need to be further discussed and explained. For instance, the decision to remove three of the five original variation operators from EoH is justified empirically.
4. The success of the reduction phase hinges on the LLM having relevant knowledge about related COPs. For truly novel or niche problems, the LLM may fail to generate meaningful reductions. This is a critical failure mode, but only briefly touched upon in the limitations.

**Questions:**

1. What was the rationale for choosing $M_{init}=10$ and $M=3$? The article would benefit from an explanation of the hyperparameter choices.
2. The "reduction refinement" step is vital. It is recommended to provide statistics from your experiments on its activation frequency. For instance, in a typical 20-generation run for TSP, how many times was a reduction refined on average?
3. What kinds of changes does the reduction refinement step typically produce? Are they minor tweaks to the implementation of $(f, g)$, or do they represent fundamentally different reduction strategies? An example would be very helpful.
4. The author removed the E1 operator (generate a completely new heuristic) from EoH. This seems counterintuitive, as RedAHD is built on exploration. It is recommended to provide a clearer explanation to this question.
5. For the CVRP results in Table 7, the performance jump when using 03-mini is dramatic.
It is suggested to provide a qualitative comparison of the reductions and/or heuristics generated by GPT-40-mini versus 03-mini for this problem. What specifically did the more capable model do better?
6. For the highly constrained VRPTW, noted that a high percentage of invalid solutions. Is this because the LLM struggles to generate valid reduction functions $(f, g)$, or because the subsequent LLM-EPS search fails to produce heuristics that respect the constraints of the reduced problem B?

---

> ### Author Response · Authors · 2025-11-22
>
> Dear Reviewer 3HNV,
>
> We greatly appreciate your insightful reviews and encouraging comments. We will try our best to address your questions and concerns as follows.
>
> ## Question 1
> Although RedAHD introduces a new set of reduction-related hyperparameters ($M_{init}=10, M=3, l=3, T=3$), we did not perform any hyperparameter tuning throughout our experiments. In fact, they were originally chosen to simply ensure reasonable resource consumption (as discussed in Appendix D.5).
>
> ## Question 2
> The activation frequency of the reduction refinement step depends on the initial pool of LRs, which can fluctuate across different runs of RedAHD (though we found this has little impact on the final performance, please refer to our newly added sensitivity analysis in Table S13 of the revised paper). In our experiments, for $T=3$, we found that this refinement step was called at least once up to three times. We have added this observation in Line 430 of the revised PDF.
>
> ## Question 3
> The reduction refinement step may perform syntactic and sometimes algorithmic tweaks to the implemented $(f,g)$. Please refer to Figure S7 in the revised PDF of our paper for an illustrative example, where we have added the original, pre-refined LR for MKP in addition to its refined version (bottom and center of Figure S7, respectively).
>
> ## Question 4
> In the early stages of developing RedAHD, we did include the E1 operator in EoH but noticed no improvement or sometimes even worse performance than when it is omitted. Upon deeper investigation, we found that the heuristics generated by E1 are more likely to be discarded during fitness evaluation (due to either syntactic errors or exceeding the evaluation time limit). One explanation could be that the multi-problem search procedure from RedAHD enabled by the pool of LRs already provides sufficient exploration of the heuristic space (as indicated by its vital role in boosting the performance of RedAHD based on Table S12), hence additional exploration via E1 becomes redundant.
>
> ## Question 5
> We attribute the performance jump of RedAHD when using o3-mini in place of GPT-4o-mini more to the reduction phase. As an example, we qualitatively compare the reduced problems between the two LLMs below.
>
> Problem B Description (o3-mini)
> > Problem B7 involves constructing a layered decision process where the first layer uses a heuristic to allocate customers to different vehicles and the second layer optimizes the sequence within each allocation using a modified TSP method.
>
> Problem B Description (GPT-4o-mini)
> > Problem B6 involves utilizing a nearest neighbor heuristic to develop initial routes, followed by a local search algorithm to refine those routes for reduced travel distance.
>
> We see that the LR designed by o3-mini is more sophisticated with a nontrivial breakdown of the original COP compared to the LR from GPT-4o-mini.
>
> ## Question 6
> For VRPTW, we found that the invalid solutions stem mainly from the LLM-EPS search failing to produce heuristics that respect the constraints of the respective problem B. More specifically, the designer LLM (GPT-4o-mini) seems to prioritize the satisfaction of vehicle capacity constraints over time window constraints, which indicates that the former is more well-studied in literature.

---

### Official Review · Reviewer_EzGv · 2025-10-31

**Soundness:** 3
**Presentation:** 3
**Contribution:** 2
**Rating:** 4
**Confidence:** 4

**Summary:**

This paper identifies a key limitation in current LLM-based Automatic Heuristic Design (AHD) methods: they are not truly end-to-end and rely on human experts to specify a Generalized Algorithmic Framework (GAF), such as ACO or IC, which reintroduces significant domain knowledge and implementation effort. To address this, the authors propose RedAHD, a novel framework that enables LLM-AHD methods to operate without a predefined GAF. The core idea is to use an LLM to automate problem reduction: the LLM is prompted to transform the target CO problem A into a similar CO problem B. The LLM generates the functions to map instances from A to B and solutions from B back to A. Existing LLM-EPS methods are then used to design heuristics for directly solving problem B, thereby indirectly solving problem A. The RedAHD framework consists of three main components: 1) Reduction Initialization: An LLM generates a set of candidate Language Reductions (LRs) and a set of heuristics; 2) Multi-Problem LLM-EPS: A novel evolutionary search where heuristics from different LRs (i.e., for different "Problem B's") can be used as references to create new heuristics for other LRs, facilitating the discovery of novel algorithmic ideas; 3) Reduction Refinement: An LLM refines the reduction functions if the search for an LR stagnates, helping to avoid local optima. The authors evaluate RedAHD on six COPs (TSP, CVRP, KP, MKP, OBPP, BPP) and show that it achieves competitive or improved results compared to SOTA LLM-AHD methods that rely on manually specified GAFs.

**Strengths:**

1. This paper's core premise is sound. Instead of automating heuristic design within a fixed framework, this work attempts to automate the framework itself by re-framing it as a problem-reduction task.

2. The paper is well-written and clearly motivated. The problem statement is easy to understand, and the high-level schematic in Figure 2 effectively communicates the method's workflow.

3. The proposed RedAHD framework is well-structured and thoughtfully designed with its three components (initialization, multi-problem search, refinement).

**Weaknesses:**

1. It appears the framework has simply shifted the manual-effort burden. Experts must still manually design detailed, COP-specific prompt components (Tables S10, S11) and, most critically, manual solution checkers. The paper's own experiment on VRPTW demonstrates this new burden is a critical point of failure. The authors state that the designed heuristics "are not consistently valid" and violate constraints in over 40% of test instances.

2. The experiments set $M_{init}=10$ and $M=3$, but provide no analysis on the quality of this initial pool. How sensitive is the final performance to this initialization step? This critical component seems under-analyzed.

3. The ablation in Table S12 shows that the multi-problem component ($M=3$) is significantly better than the single-problem component ($M=1$). This suggests much of the performance gain might come from the multi-problem search. However, the current baselines (e.g., EoH on ACO) are single-problem (or single-GAF).

**Questions:**

1. The VRPTW experiment highlights that designing the prompts and solution checks is a new, expert-level burden and a critical failure point. How do you quantify this new manual effort against the effort of implementing a GAF? Given that the method can produce invalid solutions for complex COPs, how can the claim of "enhanced automation" be justified?

2. How sensitive is RedAHD to the quality of the initial $M_{init}=10$ LRs? In your experiments, what percentage of these initial LRs were trivial or invalid? What happens if the LLM fails to generate any non-trivial strategies for a new problem?

3. How much of the performance gain over GAF-based baselines is attributable only to the GAF-free reduction aspect, versus the multi-problem search strategy?

4. The problems solved are limited to the vehicle routing and packing problems. This problem seems relatively easy to reduce. Can the proposed method be extended to handle more complex problems such as the flow shop scheduling problem (which is solved in EoH)?

---

> ### Author Response · Authors · 2025-11-22
>
> Dear Reviewer EzGv,
>
> We greatly appreciate your insightful reviews of our paper. We hope to address all your questions and concerns below.
>
> ## Question 1
> Given a COP, the manual efforts required for leveraging existing LLM-EPS works include (1) specifying a suitable GAF and identifying one of its components to be designed by LLMs, (2) implementing the GAF, (3) designing prompts pertaining to the employed GAF, and (4) building the fitness evaluation function based on the GAF. Notably, the aforementioned steps must be repeated for different COPs even when employing the same class of GAF (see Appendix C for comparison between ACO code for TSP vs. CVRP).
>
> On the other hand, being self-contained and independent of GAFs, the manual efforts from running RedAHD for a given COP only include (i) designing prompts for candidate LR generation and (ii) specifying solution checks during fitness evaluation as previously mentioned in our Limitation section (Appendix D.6). From our breakdown of the manual efforts involved, we can see that (i) and (ii) are essentially equivalent to (3) and (4), respectively. While (i) might require extra care for more challenging COPs such as VRPTW, the extent of human expert involvement is small compared to fulfilling (1) and (2). Thus, given our goal of reducing the manual efforts toward end-to-end LLM-based AHD as highlighted throughout the paper, we believe our claim of “enhanced automation” is justified.
>
> ## Question 2
> Regarding the sensitivity of the initial LRs to the final performance of RedAHD, please see Table S13 from Appendix D.2 in the revised PDF where we reran RedAHD five times for TSP. Our analysis demonstrates that the performance of RedAHD does not heavily rely on the quality of the initial LRs, which agrees with your excellent observation from Weakness 3 and Question 3 on the importance of our multi-problem search procedure (to be addressed shortly).
>
> The quality of the initial LRs depends on how knowledgeable the employed LLM is on the considered COP. For better-studied COPs such as TSP, we observed more than half ($>5>M=3$) of the initial LRs are non-trivial and produce effective heuristics. For less-studied COPs such as CVRP, we noticed a decrease in non-trivial LRs but no less than our specified $M=3$.
>
> In cases when the LLM generates insufficient non-trivial LRs, as previously mentioned in the Limitation section (Appendix D.6, where we also clarified how to detect trivial LRs), we recommend switching to a more capable LLM with adequate domain knowledge. In our experiments, our LLM of choice (GPT-4o-mini) was able to design effective LRs for all considered COPs.
>
> ## Question 3
> As you correctly pointed out, the performance gain of RedAHD over GAF-based baselines stems mainly from its multi-problem search procedure, which is enabled by the pool of reductions generated in the previous step. Without this multi-problem component, RedAHD relies solely on the encoded knowledge of the designer LLM to generate GAF-free heuristics for the respective reduced COP and hence may struggle to generate novel algorithmic ideas (Mahowald et al., 2024), resulting in subpar performance compared to GAF-based baselines. To verify this claim, from our ablation in Table S12, we ran RedAHD without the multi-problem search while keeping $M=3$ and observed that the performance is no better than RedAHD with $M=1$.
>
> Concerning Weakness 3 in your review, we would like to mention that the proposed multi-problem search procedure does not give RedAHD an unfair advantage over single-problem GAF-based baselines. Rather, it simply makes use of the vast encoded knowledge of LLMs to discover novel heuristics from uncharted heuristic space. Please let us know during the discussion phase if our response to your concern is satisfactory.
>
> ## Question 4
> Regarding our considered set of COPs in the paper, we selected them based on their prevalence in existing LLM-EPS works (i.e., considered in at least two papers as shown in Table 1) to ensure fair benchmarking.
>
> Per your suggestions, please refer to Table S14 in the revised PDF for our newly added experiments on FSSP, where RedAHD outperforms the best baseline in nearly all settings. We would like to note that in the original EoH paper, GLS was employed as the GAF. While the resulting performance is indeed excellent, employing GLS requires significantly more domain knowledge and manual efforts. From our response to Question 1, this burden mainly includes (1) understanding of GLS and the identification of one of its components to be designed by LLMs (in this case the objective landscape in GLS for guiding the local search toward more promising areas), and (2) its implementation based on (1). When IC is chosen instead as the GAF, we observe a substantial drop in EoH performance, which is well below existing baselines. Thus, our results confirm that even without relying on GAFs, RedAHD can effectively handle complex COPs beyond vehicle routing and packing problems.

---

> > ### Comment · Reviewer_EzGv · 2025-11-26
> >
> > Thank you very much for your response. Some of my concerns are addressed. However, I notice that this paper only lists the gap and obj. as the metric. Why not list the inference time? What about comparing the inference efficiency? Does the proposed method need more time for the multi-problem search procedure? What if the baselines also conduct a multi-GAF (e.g., not only GLS but also IC and so on) search procedure so as to trade more time to improve the solution quality?

---

> ### Author Response · Authors · 2025-11-28
>
> Dear Reviewer EzGv,
>
> We appreciate your follow-up questions and will try our best to clarify your concerns below.
>
> > This paper only lists the gap and obj. as the metric. Why not list the inference time? What about comparing the inference efficiency?
>
> Regarding **efficiency benchmarking**, we followed the standard evaluation protocol from existing LLM-EPS works. For this reason, as stated in Appendix D.5 from our original submission, we mainly focus on the number of fitness evaluations rather than raw training or inference time. For inference specifically (in both ID and OOD settings), RedAHD is guaranteed to share similar efficiency with existing LLM-EPS methods given that during training we maintained the same maximum running time for each instance (60 seconds).
>
> > Does the proposed method need more time for the multi-problem search procedure?
>
> We understand your thoughtful concern. As detailed from Line 300, we deliberately employed **LR ration** to prevent additional runtime from running the search with multiple LRs: “To maintain the number of newly generated heuristics in a generation, each variation operator is applied to generate only $0<N_j<N$ heuristics for $B_j$ so that $\sum^M_{j=1}Nj=N$.” This applies to both with and without the proposed multi-problem search procedure, since the only difference is for the former, any heuristic, regardless of which COP it is intended to solve, may be indiscriminately selected as parent when designing new heuristics for a given COP.
>
> > What if the baselines also conduct a multi-GAF (e.g., not only GLS but also IC and so on) search procedure so as to trade more time to improve the solution quality?
>
> For clarifications, when LLM-EPS methods are employed within a GAF, the (sub)problem delegated to LLMs is specific to the underlying GAF. Consider TSP for example, within IC, the problem involves selecting the next node to visit; whereas within GLS, the problem now involves updating the objective function (i.e., landscape) to guide the local search toward more promising areas. For this reason, input prompt components for the LLM (e.g., heuristic description and function signature in Table S9) are essentially different across GAFs (e.g., IC vs. GLS), rendering heuristics designed for a specific GAF inapplicable to another GAF. Therefore, from a practical standpoint, we believe it is not possible to conduct a multi-GAF search procedure for existing LLM-EPS baselines. (We believe the confusion may stem from the oversimplification of the demonstration in Figure 3, and we will make appropriate clarifications if necessary.)

---

### Official Review · Reviewer_2KLT · 2025-11-01

**Soundness:** 2
**Presentation:** 2
**Contribution:** 2
**Rating:** 4
**Confidence:** 4

**Summary:**

This paper proposes an augmentation method called reduction for building LLM-based AHD methods. Generally, RedAHD can achieve impressive results on TSP. I served as the reviewer for this paper at previous conferences. After comparing the changes, I tend to barely retain my previous review

**Strengths:**

1. The introduction part is well-written. With clear evidence and good logic, clear improvements compared to the previous manuscripts.

2. The proposed RedAHD shows impressive results on TSP and some other COPs.

**Weaknesses:**

See Questions

**Questions:**

1. RedAHD seems to have significant differences in performance on different issues. According to Figure 3, RedAHD is able to design a 2-opt operator in the TSP problem, which seems to be something that the IC framework cannot achieve. Is this the only reason why RedAHD performs well on TSP? Can performing 2-opt post-processing on the solution of EoH artificially achieve similar performances to RedAHD?

2. Can RedAHD show potential on some problems highly requiring an end-to-end heuristic (e.g., maybe TSPTW, which has no feasibility guarantee for the IC framework, and there are no current implementations with ACO)?

---

> ### Author Response · Authors · 2025-11-18
>
> Dear Reviewer 2KLT,
>
> We appreciate you taking the time to review our paper again.
>
> First off, we would like to summarize the changes we have made in our paper based on your suggestions from the previous conference.
>
> **Clarity in nomenclature and our contributions**. Concerning the use of “reduction” throughout the paper, which might be ambiguous to readers, we ensured our definition of “(language) reduction” is crystal-clear (Definition 2). Furthermore, following your suggestions on focusing more on practical implementation instead of concept, we tried our best to highlight the scope of our work by adding a summary of the advantages of RedAHD in practice (Appendix D.7).
>
> Next, we address your two raised questions, which we have been actively addressing since your previous review.
>
> ## Question 1
> Regarding your comments on **2-opt post-processing**, we believe the good performance of RedAHD, particularly for TSP, stemmed not only from its ability to leverage 2-opt operators. In fact, as shown in Figure S9 in Appendix C.4, the best performing heuristic for TSP using RedAHD does not require the use of 2-opt operators. For further evidence, we have run 2-opt post-processing on the solutions of EoH (under ACO) per your suggestions and found in the table below that RedAHD still yields significantly better performance on OOD settings (n=100).
>
> |n (number of nodes)|50|100|
> |-----------|-----------|-----------|
> |Best baseline|5.801|8.179|
> |EoH|5.828|8.263|
> |EoH (with 2-opt post-processing)|5.812|8.168|
> |RedAHD[EoH]|5.819|**8.039**|
>
> (Truncated results from Table 5 are used as references. Values capture tour lengths, hence lower values are better.)
>
> ## Question 2
> Following your helpful suggestions regarding **TSPTW**, we included an additional experiment in Appendix D.6 to speculate the potential use of RedAHD on the vehicle routing problem with time windows (VRPTW), which is more general than TSPTW (and hence serves to be a more challenging COP) and also motivated by the fact that there have been recent attempts to apply LLM-EPS methods within IC for tackling VRPTW (from the Python library developed by Liu et al. 2024c). Our results show that the heuristics designed by RedAHD may not return consistently valid solutions, which is likely due to the confined heuristic space stemming from the time window constraints. When we relaxed the constraints by allowing vehicles to fulfill customers’ demands early, we observed a significant improvement in solution validity. We acknowledge this is the current limitation of RedAHD and have revised the Limitation section accordingly (Appendix D.6).
>
> If it would help improve your evaluations on RedAHD, we are happy to provide our currently developing workaround that improves RedAHD’s effectiveness on VRPTW/TSPTW or try other COPs. In short, our proposed workaround involves more careful design of the LLM prompts used during the LR generation process in order to guarantee feasibility.

---

> ### Author Response · Authors · 2025-11-24
>
> Dear Reviewer 2KLT,
>
> As the end of the discussion phase approaches, we would like to check in with you on whether we have addressed all your raised concerns. If not, please let us know and we will try our best to provide a timely response.
>
> Sincerely,
>
> Authors of Submission 20169

---

> > ### Comment · Reviewer_2KLT · 2025-11-24
> >
> > Dear Authors,
> >
> > Thank you so much for your prompt response and follow-up.
> >
> > As a marginal paper, I acknowledge there are notable refinements in this version, especially in clarity and claims.
> >
> > However, some of my claims are not solved yet. For Question 1, according to Figure S9, it is clear that the algorithm found by Red AHD is the most basic implementation of 2-opt. This has nothing to do with whether other algorithms with 2-opt are worse or not. If we want to demonstrate superiority, we should forcibly prohibit the model from adopting 2-opt (such as limiting complexity and runtime)
> >
> > Overall, after careful consideration, I still do not believe that the text meets the admission criteria. The reasons are as follows.
> >
> > 1. The idea presented in this article is not intuitive. In fact, if you want to achieve heuristic design without LLM-EPS, this article should highlight generality and solve the obscure problem that cannot be covered by a simple framework. However, for issues that are too obscure, LLM may not involve good reduction, resulting in failure, which is somewhat contradictory.
> >
> > 2. Many experimental results in this article are confusing. This article is dedicated to showcasing relatively poor results, honest but surprising, and unable to grasp strengths and weaknesses. The author can fully consider reproducing the baseline algorithm in future submissions.
> >
> > 3. This article lacks a grasp of the significance of reduction. I once suggested renaming it as language augmentation, but I believe that the reduction claimed in this article is only a macroscopic perturbation, which does not have a logical advantage (perturbing more in a limited search may lead to poorer results).
> >
> > I am super positive to see your defense.
> >
> > Best, Reviewer.

---

### Meta-Review · Area_Chair_XLa2 · 2025-12-24

**Summary:**

Across the three reviewers, the main concerns converge on limited demonstrated advantage, fairness of comparison and true end-to-end automation. While reviewer 3HNV is positive about this work, reviewers 2KLT and EzGv question whether RedAHD’s empirical gains (especially on TSP) are fundamentally driven by the discovery of well-known operators such as 2-opt, and whether similar performance could be achieved by simply augmenting existing LLM-EPS baselines with post-processing. They also raise fairness issues in experimental comparison. Particularly, they note that RedAHD is self-contained while baselines rely on IC or ACO frameworks with different expressive power, which may bias results. In addition, EzGv points out that RedAHD may shift rather than eliminate manual effort, since COP-specific prompts, solution checkers and feasibility handling (e.g., VRPTW validity failures) still require expert intervention, which may undermine the “enhanced automation” claim.

From my own perspective, if the authors would like to justify the advantage of their end2end (E2E) approach over the non-E2E ones, they may focus on problems that non-E2E cannot handle well, as from the experiment results we can tell, the end2end approach is not significantly better, on problems that can be solved by non-E2E. On the other hand, I also have the concern in the fairness about comparing IC framework with the ones which can use 2-Opt (or similar ones). Moreover, the reduction idea does have the merit, but it might be limited in generalization, especially to hard problems with complex constraints, which were not fully justified.

**Reviewer Concerns:**

From my perspective, most of reviewer 3HNV’s concerns, particularly those related to clarity of contribution, formalization of “language reduction” and the high-level motivation of end-to-end AHD, were addressed. However, reviewer 2KLT remained unconvinced on whether the improvement is fundamentally significant or whether restricting common operators (e.g., 2-opt) would erase RedAHD’s advantage. For reviewer EzGv, the rebuttal clarified the intent of “enhanced automation” and acknowledged limitations on feasibility and manual components, but key concerns may remain outstanding, especially the argument that RedAHD shifts rather than removes expert burden (prompt engineering and solution checking) and that performance gains may stem more from the multi-problem evolutionary search than from reduction itself. Consequently, some negative concerns persist, although the authors made massive rebuttal and have resolved some other concerns.

**Reviewer Scores:**

Based on my understanding, I would say the final score remained to be 4, 4, 8.

---

### Decision · Program_Chairs · 2026-01-26

Reject